# Long-term adult human brain slice cultures as a model system to study human CNS circuitry and disease

Niklas Schwarz[1], Betül Uysal[1], Marc Welzer[2,3], Jacqueline C Bahr[1], Nikolas Layer[1], Heidi Löffler[1], Kornelijus Stanaitis[1], Harshad PA[1], Yvonne G Weber[1,4], Ulrike BS Hedrich[1], Jürgen B Honegger[4], Angelos Skodras[2,3], Albert J Becker[5], Thomas V Wuttke[1,4†*], Henner Koch[1†*]

[1]Department of Neurology and Epileptology, Hertie-Institute for Clinical Brain Research, University of Tübingen, Tübingen, Germany; [2]Department of Cellular Neurology, Hertie-Institute for Clinical Brain Research, University of Tübingen, Tübingen, Germany; [3]German Center for Neurodegenerative Diseases (DZNE), Tübingen, Germany; [4]Department of Neurosurgery, University of Tübingen, Tübingen, Germany; [5]Department of Neuropathology, Section for Translational Epilepsy Research, University Bonn Medical Center, Bonn, Germany

**Abstract** Most of our knowledge on human CNS circuitry and related disorders originates from model organisms. How well such data translate to the human CNS remains largely to be determined. Human brain slice cultures derived from neurosurgical resections may offer novel avenues to approach this translational gap. We now demonstrate robust preservation of the complex neuronal cytoarchitecture and electrophysiological properties of human pyramidal neurons in long-term brain slice cultures. Further experiments delineate the optimal conditions for efficient viral transduction of cultures, enabling 'high throughput' fluorescence-mediated 3D reconstruction of genetically targeted neurons at comparable quality to state-of-the-art biocytin fillings, and demonstrate feasibility of long term live cell imaging of human cells *in vitro*. This model system has implications toward a broad spectrum of translational studies, regarding the validation of data obtained in non-human model systems, for therapeutic screening and genetic dissection of human CNS circuitry.

DOI: https://doi.org/10.7554/eLife.48417.001

**\*For correspondence:**
Thomas.Wuttke@med.uni-tuebingen.de (TVW); henner.koch@uni-tuebingen.de (HK)

†These authors contributed equally to this work

**Competing interests:** The authors declare that no competing interests exist.

## Introduction

The human brain is composed of an intricate network of diverse cell types with complex interactions and connections (*DeFelipe et al., 2002*; *Somogyi et al., 1998*). The microstructure within the brain was already appreciated in the early pioneering work by Santiago Ramón y Cajal and has been further investigated in the last decades in post mortem human brain samples (*Elston and DeFelipe, 2002*; *Elston et al., 2001*) and in numerous animal models (*Branco and Häusser, 2011*; *Somogyi et al., 1998*). Neurons function as a core component of the microstructure of CNS circuits. The computation and integration of signals by these cells and the output is believed to underlie higher brain functions such as cognition and learning. Two broad classes of neurons, glutamatergic neurons (pyramidal neurons, 80%) and GABAergic interneurons (20%), populate the human neocortex. Cortical pyramidal neurons can be further categorized into four classes according to their output connectivity. Intratelencephalic pyramidal neurons are distributed throughout cortical layers 2–6 and build associative connections with other cortical areas (within one hemisphere or across the corpus callosum) or with the striatum. Pyramidal tract neurons reside in layer five and project their axons to

the brain stem and spinal cord. Corticothalamic neurons establish axonal output connectivity from layer six to area related thalamic nuclei. The fourth category comprises layer four short-axon intratelencephalic neurons, which locally relay thalamic input (*Shepherd and Rowe, 2017*). With the exception of layer four intratelencephalic neurons in sensory areas (spiny stellate cells) pyramidal neurons typically show a long apical dendrite. Overall morphology and complexity of apical and basal dendrites can substantially differ between pyramidal neurons and throughout layers. Furthermore, work by colleagues in the field provides direct evidence that there also exist marked species-specific differences of structural features and functional properties of pyramidal neurons (*Mohan et al., 2015*) as well as regarding basic cortical properties (*Eyal et al., 2016*), cell types (*Boldog et al., 2018*; *Wang et al., 2015*), morphology (*Elston et al., 2001*; *Mohan et al., 2015*), functional divergence (*Napoli and Obeid, 2016*) and plasticity mechanisms (*Verhoog et al., 2013*). Such differences underscore the challenges of modeling human disease by cellular and animal models and call for the development of new translational approaches based on human nervous tissue. Spare human CNS tissue obtained from neurosurgical procedures has proven valuable throughout the years, for example for electrophysiological studies in acute slice preparations (*Bernard et al., 2004*; *Kalmbach et al., 2018*; *Kerkhofs et al., 2017*). Unlike with rodent CNS tissue, past attempts by different groups to establish organotypic slice cultures of human CNS tissue had yet repeatedly faced pronounced limitations regarding viability and integrity (*O'Connor et al., 1997*; *Verwer et al., 2002*). However, recent efforts challenged these limitations and provided promising proof-of-concept data on electrophysiological function (*Eugène et al., 2014*) and optogenetic targeting (*Andersson et al., 2016*) in human brain slice cultures. Subsequent studies established successful neuronal labeling, optical manipulation and calcium imaging by virus-driven rapid expression of respective probes in short term cultures of several days (*Ting et al., 2018*). Recent parallel work by us in contrast focused on optimization of culture conditions and demonstrated that neuronal viability and network activity of human brain slice cultures can be significantly extended to up to three weeks under specified conditions (*Schwarz et al., 2017*). Proper function of complex CNS networks, however, critically depends on morphological and functional integrity of neurons structurally underlying circuitry. Neurons need to extend their dendritic processes to correct locations in order to receive appropriate synaptic inputs and relay them to adequate target structures. Specific functional properties are required to compute and integrate information in a meaningful way. Whether this level of morphological complexity along with inherent electrophysiological characteristics is maintained in human brain slice cultures over a period of two to three weeks or whether dendritic arbors rarefy and functional features dissipate is not known and will be investigated in this work.

In the last decades, important molecular genetic achievements and subsequent combined application of tools such as the Cre-lox system (*Song and Palmiter, 2018*), optogenetics (*Kim et al., 2017*; *Madisen et al., 2012*) and virus-mediated gene targeting/knockout/delivery (*Hendrie and Russell, 2005*; *Howard et al., 2008*; *Rao and Craig, 1997*) enabled an increasingly better understanding of how distinct cell types assemble and form functional circuitry in the rodent brain. Adapting such strategies to human brain slice cultures reflecting an environment as close as possible to the actual human brain will open new roads toward dissection of human CNS circuitry and could potentially provide an increasingly more accurate understanding of how human CNS disease develops.

Toward these aims, we now performed detailed cellular analyses of the somatodendritic and synaptic spine compartments and of characteristic electrophysiological properties of single neurons throughout the course of culturing. These features form the basis of intact network function but whether they are maintained in cortical slice cultures remained still elusive. Further experiments established the conditions for efficient viral transduction with a focus on glutamatergic pyramidal neurons within human brain slice cultures.

We find robust structural and electrophysiological stability of human pyramidal neurons and amenability of human brain slice cultures to efficient genetic manipulation, as revealed by successful transduction by GFP-encoding adeno-associated viral vectors (AAV-vectors). GFP fluorescence levels were sufficient for both post-hoc confocal microscopy- and two-photon live cell imaging-based assessment of neuronal morphology including spines emanating from the dendritic arbors. We demonstrate GFP-based cellular 3D morphological analysis with increased efficiency and compatible quality in comparison to classic intracellular biocytin fillings.

The presented data indicate feasibility of utilizing human brain slice cultures as a model system as close as possible to the actual human brain and apply them toward studies of human CNS circuitry, disease and therapeutic screening, thereby having the potential to close the translational gap to non-human model systems.

## Results

### Electrophysiological properties of pyramidal neurons in human brain slice cultures versus acute slices

Besides morphological and structural features intact physiology of neurons constitutes one of the hallmarks of healthy neural tissue. In a first set of experiments we determined the stability of the electrophysiological properties of pyramidal neurons performing whole cell patch clamp recordings (n = 45) from cells in acute (n = 17) or cultured (n = 28) slices (2–14 DIV). The cells recorded from cultured slices were further subdivided into early (2-3 DIV, n = 8) and late in culture (7–14 DIV, n = 20) for statistical group analysis (*Figure 1E*). Recordings of pyramidal neurons showed a typical regular spiking pattern in acute and cultured slices (*Figure 1B,C*), as described in previous studies investigating human pyramidal neurons in acute slices (*Verhoog et al., 2013*). The firing frequency in response to current injections was not significantly different between the cells recorded on the day of the surgery (acute slices = 0 DIV, with a peak firing rate of $50.84 \pm 9.90$ Hz in response to a current injection of 200 pA) and the cells in cultures between 2 and 3 DIV (peak firing rate: $59.4 \pm 12.46$) and 7–14 DIV (peak firing rate of $55.79 \pm 9.34$ Hz), (Kruskal-Wallis test, p=0.74, *Figure 1C*). Similarly, mean values of AP half width, sag potential amplitude and input resistance did not differ between the tested groups (0 DIV, 2–3 DIV and 7–14 DIV, Kruskal-Wallis test, p>0.05, *Figure 1E*). This was also reflected by no significant linear regression correlation of these parameters versus the DIV (*Figure 1D*).

For mean values of resting membrane potential we found a small significant difference (Kruskal-Wallis test, p<0.05) for the acutely measured cells (0 DIV, $-77.94 \pm 1.5$ mV) in comparison to early cultured (2–3 DIV, $-72.38 \pm 0.96$ mV, Dunn's multiple comparisons test, *p=0.02) and late in culture cells (7-14 DIV, $71.95 \pm 1.43$, Dunn's multiple comparisons test, *p=0.01). However, there was no significant difference between early (2–3 DIV) and late in culture (7–14 DIV) measured cells (Dunn's multiple comparisons test, p>0.99). This was also reflected by no significant linear regression correlation of the resting potential versus the DIV for cells of all three groups (*Figure 1D*). In summary, increasing time in culture does not impact the majority of analyzed intrinsic electrophysiological characteristics of pyramidal neurons, except for the membrane resting potential which slightly changed to more depolarized values within the first 2–3 days in culture and then stayed stable over the remaining time.

### Maintenance of the glutamatergic neuron population in human brain slice cultures over time

In a further set of experiments, we determined the degree of survival of the excitatory neuronal population in human brain slice cultures over time. Satb2 is a transcription factor that is expressed exclusively in excitatory neurons in adult mouse and human cortex (*Britanova et al., 2008*; *Huang et al., 2013*; *Hodge et al., 2019*; http://celltypes.brain-map.org/rnaseq). Taking advantage of Map2, a neuron-specific cytoskeletal protein, in combination with Satb2 (*Figure 1—figure supplement 1A*) we found no significant changes of the absolute numbers of double-positive neurons in slice cultures between 0 DIV (3 slices), 9 DIV (3 slices) and 14 DIV (2 slices) (Dunn's multiple comparisons test, *Figure 1—figure supplement 1B*). Double positive neurons were counted in small z-stack projections of confocal images of four to six different regions (each area 290 μm x 290 μm in size) in layers 2/3 of each slice: the absolute number of Map2 and Satb2 double positive neurons per 100 μm $\times$ 100 μm (10000 $\mu m^2$) was on average $8.57 \pm 0.60$ in acute slices (18 analyzed areas), $10.14 \pm 3.52$ (16 analyzed areas) in slices analyzed at 9 DIV and $9.49 \pm 1.77$ (9 analyzed areas) in slices obtained at 14 DIV (*Figure 1—figure supplement 1B*). To investigate whether the ratio of glutamatergic to GABAergic neurons remained stable we calculated the ratio of Satb2 and Map2 double-positive neurons to all Map2 positive neurons. Ratios were $0.72 \pm 0.02$ in acute slices, $0.59 \pm 0.09$ at 9 DIV and $0.65 \pm 0.11$ at 14 DIV and revealed a discretely lower, albeit statistically significant, ratio at 9 DIV compared to 0

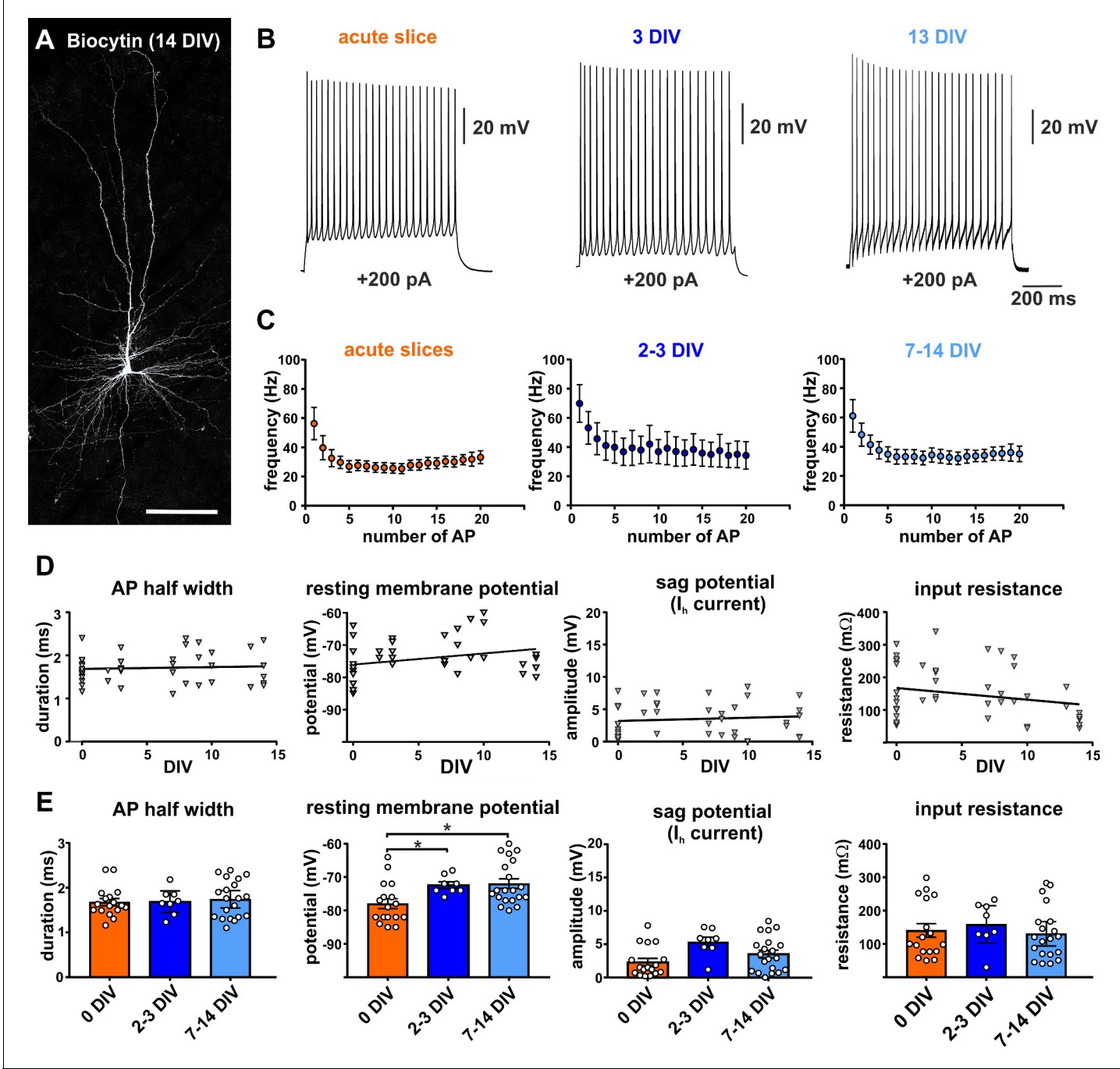

**Figure 1.** Electrophysiology of adult human pyramidal neurons. (A) Example of biocytin filled pyramidal neuron (14 DIV) after streptavidin-Cy3 counterstaining, scale bar 150 μm. (B) Typical regular spiking pattern of human pyramidal neurons (acute slice, 3 DIV and 13 DIV) in response to +200 pA positive current injection. (C) Neurons recorded in acute slices (within 12 hr after surgery) and in brain slice cultures (2–3 DIV and 7–14 DIV) showed similar action potential firing frequencies and typical spike frequency adaptation upon 200 pA current injection. (D, E) Quantification of (from left to right) the AP half width, resting membrane potential, sag potential amplitude and input resistance revealed no significant correlation between these values and the days *in vitro* (DIV), (E) except membrane resting potential which slightly changed to more depolarized values within the first 2–3 days in culture and then stayed stable over the remaining time.

DOI: https://doi.org/10.7554/eLife.48417.002

The following figure supplements are available for figure 1:

**Figure supplement 1.** Satb2 positive neurons in human brain slice cultures.

DOI: https://doi.org/10.7554/eLife.48417.004

*Figure 1 continued on next page*

*Figure 1 continued*

**Figure supplement 2.** Stability of gross structural features of cortical slices between surgeries and over time in culture.
DOI: https://doi.org/10.7554/eLife.48417.003

DIV, but no difference between 0 DIV and 14 DIV or 9 DIV compared to 14 DIV (Dunn's multiple comparisons test; *Figure 1—figure supplement 1B*).

These data indicate robust survival of excitatory and inhibitory neurons with a sustained ratio of excitatory to inhibitory neurons throughout the culturing time.

## Viral transduction of human brain slice cultures

To investigate whether neurons in human brain slice cultures can be genetically targeted with viral vectors and whether such strategies could render morphological analysis of human neurons more efficient than approaches solely relying on classic biocytin fillings, we used retrograde adeno-associated virus (AAVrg) encoding GFP under the human synapsin promoter to specifically transduce neurons. The slices were injected with the virus at 3–5 DIV using a picospritzer, cells were then recorded and imaged at 8–16 DIV (n = 20 slices). We found robust virus transduction in all injected slices with variable expression in distinct types of neurons throughout all cortical layers (*Figure 2*). The GFP expression was clearly present in somata, dendrites including spines and axons of transduced neurons (*Figure 2A1 and A2*).

GFP expression levels were found to be very robust, suggesting that several types of analyses including quantification of whole cell morphology, spine density, spine head diameter and spine length (see below) could be achieved based on GFP fluorescence. Furthermore, viral transduction of human neurons will enable direct investigation of the impact of disease causing mutations on neuronal function and morphology as well as on neuronal networks and will enable studies of human CNS circuitry involving optogenetic and chemogenetic tools.

## Dendritic morphology of pyramidal neurons *in vitro*

Morphologically intact neurons are the basic framework of healthy neural brain tissue and degradation of dendritic arbors may even occur before an impairment of electrophysiological properties. Therefore, we set out to determine whether neurons are subject to morphological changes throughout their time in culture and performed 3D reconstructions of biocytin filled neurons at different time points between 0 DIV and 14 DIV. High-resolution confocal z-stack tile scans were acquired (*Figure 1A*), stitched with ImageJ and neurons were digitally reconstructed using Imaris software (see Materials and methods for details). Pyramidal neurons were filled with biocytin during electrophysiological recordings and 24 fillings were deemed to be of sufficient quality (clear presence of both apical and basal dendritic compartments, *Figure 1A*, while incompletely filled neurons were excluded from the analysis) to enable high-resolution morphological analysis (*Figure 3*). Reconstructed neurons were classified by the distance of their soma to the pia as layers 2/3 (distance to pia: 300–1200 µm, n = 16), layer 4 (distance to pia: 1200–1500 µm, n = 3) or layers 5/6 (distance to pia: 1500–2900 µm, n = 5) pyramidal neurons (*Figure 3*) (*Mohan et al., 2015*; *Ting et al., 2018*; *Goriounova et al., 2018*). All neurons presented a typical pyramidal neuronal shape with distinct apical dendrites (*Figure 3*, red), extensive basal dendritic trees (*Figure 3*, blue), axons (*Figure 3*, magenta) and spines present on the apical and basal dendrites. The total length of the apical and basal dendrites as well as the combined total length of the reconstructed dendritic filaments were quantified for each analyzed neuron and correlated to the distance of the respective soma to the pia and to the DIV (Figure 5A). To investigate whether neurons undergo degenerative processes or exhibit layer-dependent morphological differences, we calculated a linear regression for the values of total basal, total apical and total combined dendritic length of all neurons to the DIV and the distance to the pia. There was no significant correlation between the total basal, apical or combined dendritic length and the number of days *in vitro* for these analyzed biocytin filled neurons (n = 24, linear regression, p=0.07). While there was also no significant correlation of all three parameters of total dendritic length with overall soma distance to the pia (n = 24), additional subgroup analysis of superficially and incrementally deeper situated layers 2/3 neurons in acute and cultured cortical

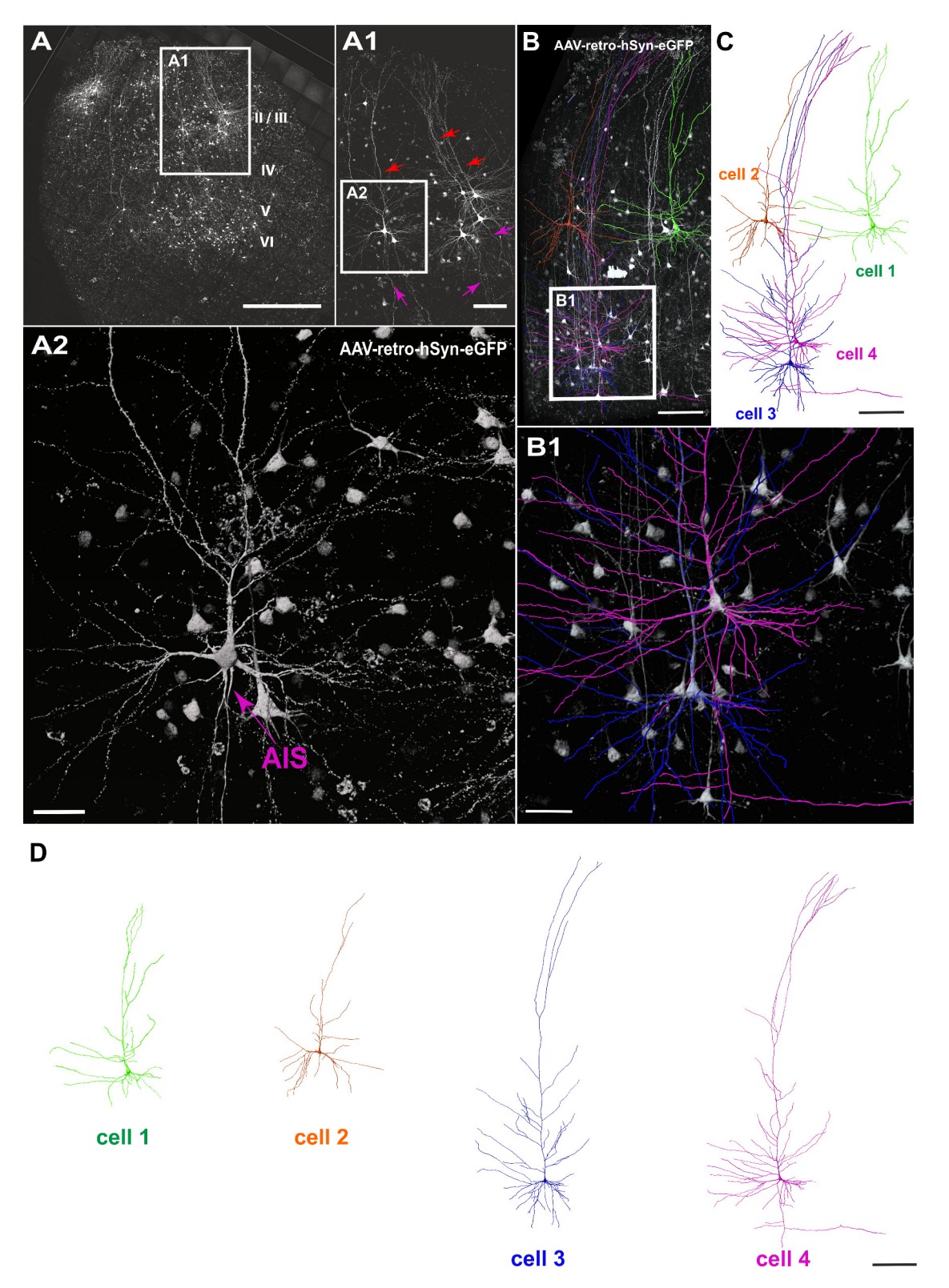

**Figure 2.** Viral transduction in human brain slice cultures and 3D reconstruction of GFP-labeled pyramidal neurons. (**A**) Representative example of a human brain slice after viral transduction with AAVrg-hSyn-GFP at 9 DIV, scale bar 1000 µm. (**A1**) Enlarged confocal image from A: layers 2/3 pyramidal neurons with intact apical dendrites (red arrows) and axons (magenta arrows), scale bar 200 µm. (**A2**) The soma and axon initial segment (AIS) are clearly visible in the virally transduced neurons, scale bar 50 µm. (**B**) 3D reconstructions of four GFP transduced pyramidal neurons were performed from

*Figure 2 continued on next page*

*Figure 2 continued*

confocal z-stack tile scans, scale bar 200 µm. The cells were individually traced and pseudo colored (C). (B1) Example of two neurons within close proximity of each other, which could still be clearly separated for further analysis, scale bar 50 µm. (D) Separation of the four distinct GFP-labeled pyramidal cells for further analysis, scale bar 200 µm.

DOI: https://doi.org/10.7554/eLife.48417.005

tissue revealed a gradually increasing total length of apical and basal dendrites (Figure 5A; n = 16), as has been described in a previous study by *Mohan et al. (2015)* in non-cultured CNS tissue.

Next, we asked whether the native GFP signal following viral transduction of cultures could be readily used for structural analysis of groups of neurons (*Figure 2B*). A typical example of reconstruction and separation of GFP-labeled neurons fur further analysis is shown in *Figure 2*. We picked a total of 23 neurons from four virally transduced slices (8, 9, 10 and 14 DIV) with strong GFP expression in layers 2–6 (*Figure 2A*). Similar to biocytin filled neurons all 23 GFP positive neurons were classified by the distance of their soma to the pia as layers 2/3 (distance to pia: 300–1200 µm, n = 12), layer 4 (distance to pia: 1200–1500 µm, n = 3) or layers 5/6 (distance to pia: 1500–2900 µm, n = 8) pyramidal neurons (*Figure 4*; apical dendrites, red; basal dendrites, blue; axons, magenta). In comparison to biocytin fillings GFP-mediated tracing equally successfully detected the gradually increasing dendritic length of layers 2/3 neurons depending on their soma position (*Figure 5B*). There was no significant correlation between the combined dendritic length and the DIV, neither for GFP-based tracings (data not shown) nor for analysis after pooling all biocytin and GFP reconstructed neurons (*Figure 5C*, middle panel). Comparing combined total dendritic length of all biocytin and GFP-labeled neurons independent of the DIV revealed a slight underestimation of values by GFP (*Figure 5C*, right panel, $p<0.01$, n = 24 (biocytin), n = 23 (GFP), unpaired Mann Whitney test) – likely because very distal dendritic ramifications are being captured somewhat less reliably (due to intermingling GFP positive processes from neighboring neurons and some arising uncertainty when judging whether to assign GFP positive processes in the periphery of the neuron being reconstructed to the filament of this neuron or whether to discard them by deeming them as originating from neighboring neurons) – but overall successful reconstruction of the major parts of dendritic arbors (*Figure 2* and *Figure 4*).

In summary a total of 47 biocytin and GFP-labeled neurons have been analyzed (*Figures 3–5*), revealing remarkable structural preservation of pyramidal neurons without clear signs of declining morphological complexity. These data not only indicate absence of widespread progressing neuronal degeneration throughout the time in culture but also demonstrate genetic labeling as a valid alternative approach to 3D morphological analysis of adult human neurons with significantly increased efficiency in comparison to classic single cell biocytin fillings.

## Electrophysiological properties of interneurons in human brain slice cultures versus acute slices

To further investigate beyond our earlier indirect assessment based on the ratio of Map2 and Satb2 double-positive neurons to all Map2 positive neurons, whether also interneurons survive in human brain slice cultures we first performed immunocytochemical stainings for calretinin. Unlike for other interneuron subtype identifiers, we were able to obtain reliable stainings for calretinin in acute and cultured human brain slice tissue. Qualitative assessment at 0 DIV and at 9 DIV demonstrated survival of at least this subset of interneurons (*Figure 6C*). Next, we performed intracellular recordings of 22 neurons that were identified morphologically or electrophysiologically as interneurons (*Figure 6A,B,D*). In comparison to pyramidal neurons interneurons represent a morphologically and electrophysiologically even more diverse class of neurons. While morphology and firing behavior are known to substantially differ between various subclasses of interneurons, there are some intrinsic properties (such as resting membrane potential, input resistance, AP half width and sag potential; *Figure 6F–G*) which can be considered comparatively more uniform and which therefore were found suitable for an assessment over time in culture. For all these parameters we did not find significant differences between neurons measured in acute (n = 7) and in cultured slices at 7–14 DIV (n = 15) (*Figure 6D–G*). The firing behavior of the recorded interneurons showed distinct properties as

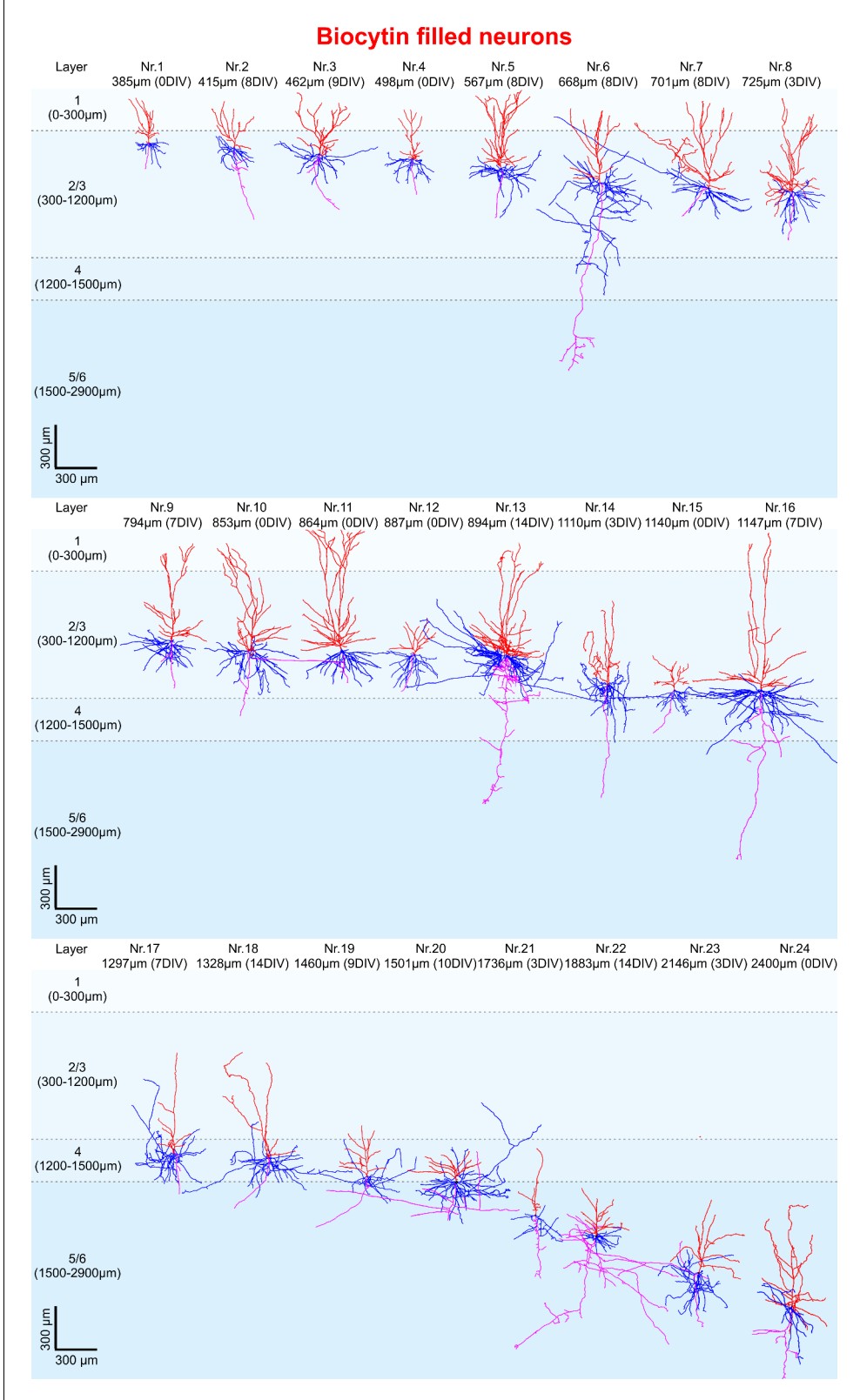

**Figure 3.** 3D reconstruction of biocytin filled pyramidal neurons. All 24 biocytin labeled reconstructed pyramidal neurons sorted by their soma distance to the pia ranging from 385 to 2400 µm represented in the colors red (apical dendrites), blue (basal dendrites) and magenta (axons).

DOI: https://doi.org/10.7554/eLife.48417.006

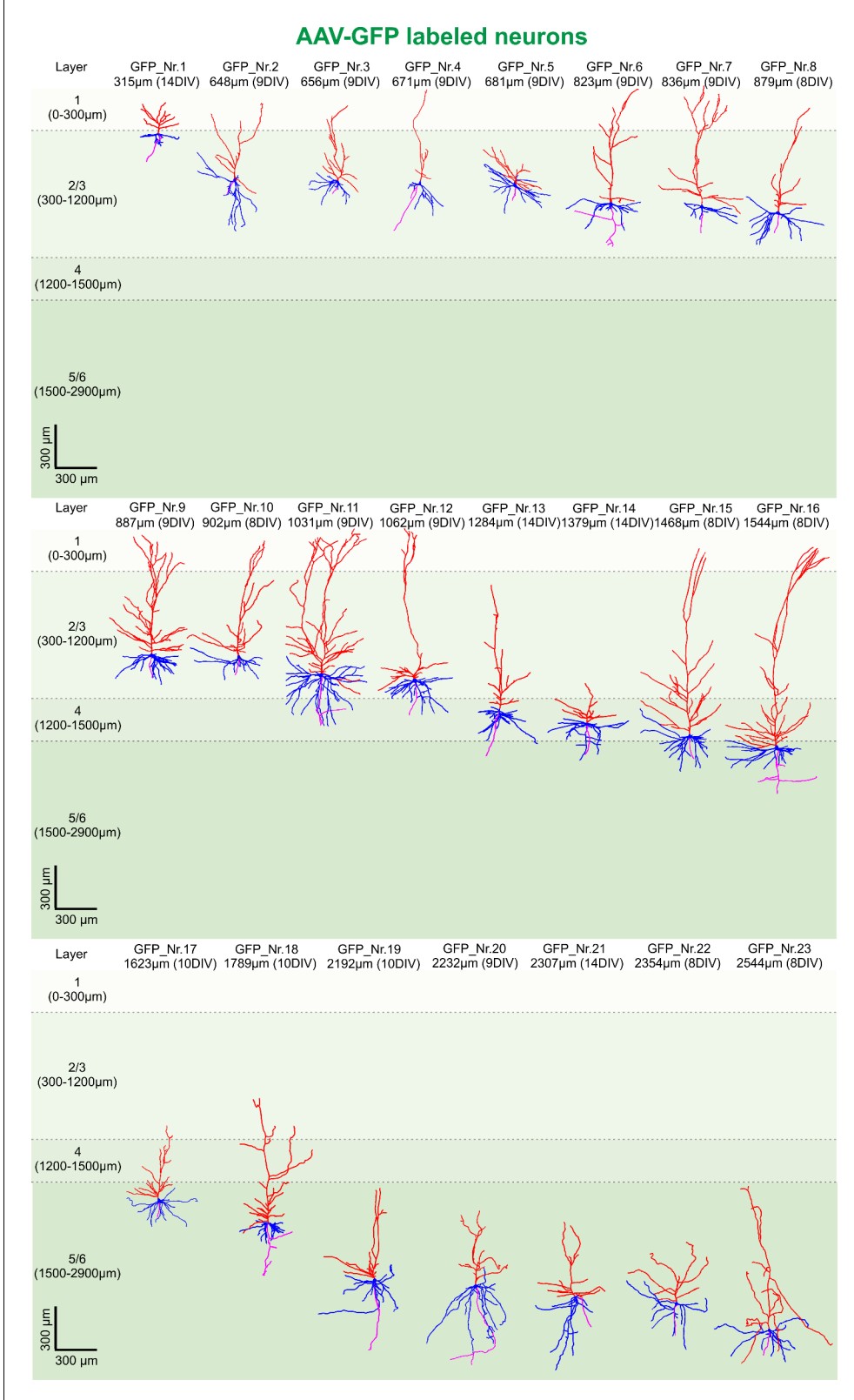

**Figure 4.** 3D reconstruction of GFP-labeled pyramidal neurons. All 23 GFP-labeled reconstructed pyramidal neurons sorted by their soma distance to the pia ranging from 315 to 2544 μm represented in the colors red (apical dendrites), blue (basal dendrites) and magenta (axons).
DOI: https://doi.org/10.7554/eLife.48417.007

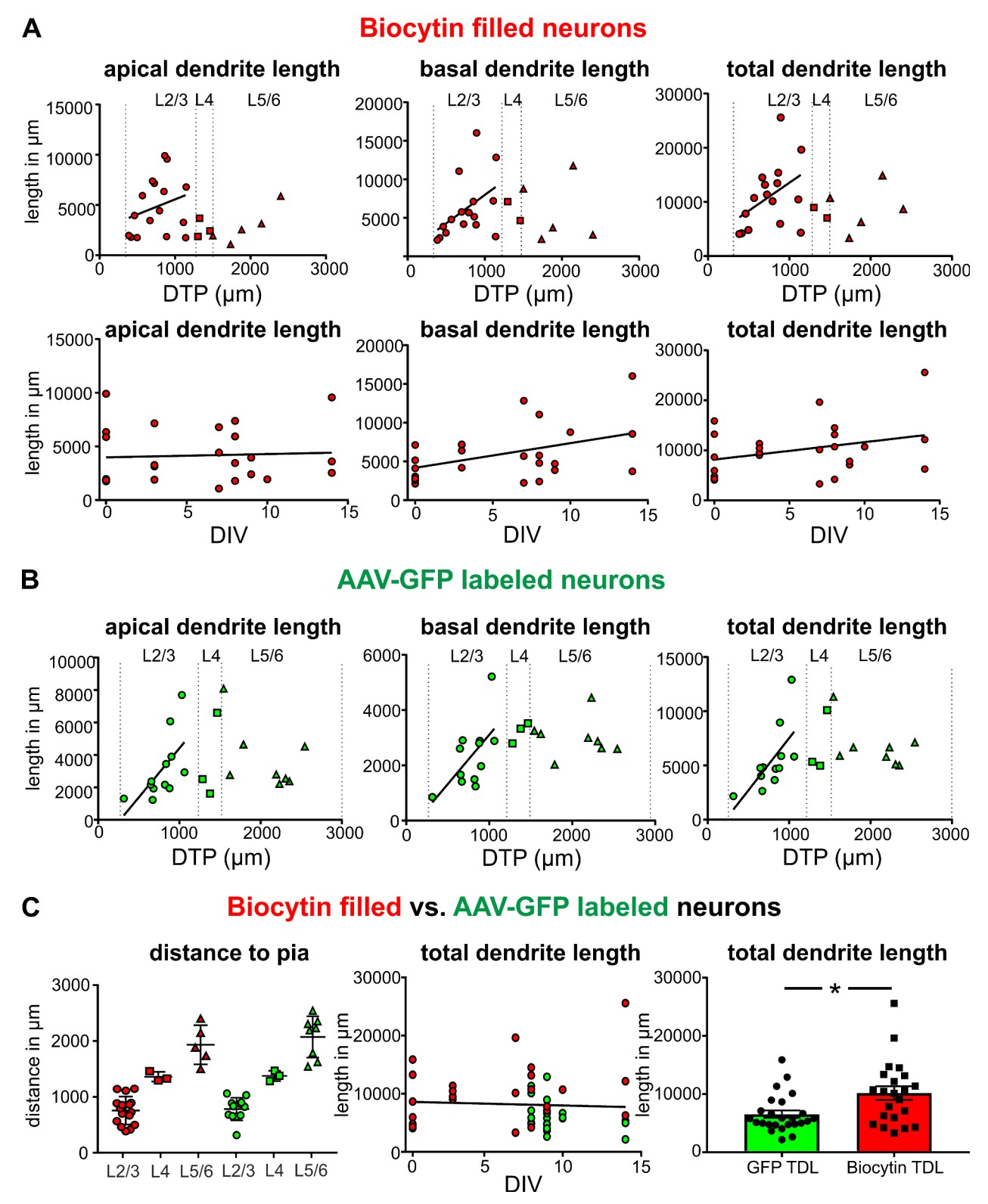

**Figure 5.** Quantification of the total length of apical and basal dendrites of biocytin filled and GFP-labeled neurons. (**A**) Upper panels: Neurons that were patched and filled in layers 2/3 are represented by red circles, neurons in layer four by red squares and neurons in layers 5/6 by red upward triangles. Bottom panels: Each biocytin filled neuron is represented by a red circle. (**B**) GFP-labeled layers 2/3 neurons are represented by green circles, neurons in layer four by green squares and neurons in layers 5/6 by green upward triangles. (**C**) Left: Analysis of the distribution of biocytin filled (red)

*Figure 5 continued on next page*

*Figure 5 continued*
and GFP-labeled (green) pyramidal neurons in the different layers. (**C**) Middle: Total dendritic length of biocytin filled (red) and GFP-labeled (green) pyramidal neurons plotted against DIV. (**C**) Right: Reconstructions of pyramidal neurons based on GFP expression slightly underestimated the total dendritic length compared to classic biocytin fillings (n = 22 for biocytin, n = 23 for GFP-labeled, **p<0.01).
DOI: https://doi.org/10.7554/eLife.48417.008

described before for rodents and humans and ranged from fast spiking (*Figure 6D*) to none fast spiking firing patterns (*Ascoli et al., 2008*).

## Spine morphology of biocytin filled and GFP-labeled pyramidal neurons in acute and cultured slices

The spines of pyramidal neurons are the presumed site of excitatory synapses and have been shown to be a plastic part of the morphology of these cells (*Trachtenberg et al., 2002*; *Yuste and Bonhoeffer, 2004*). To analyze morphological features of spines we acquired high-resolution confocal scans of dendritic compartments of biocytin filled (*Figure 7A,B*) or GFP positive (*Figure 7A,C*) pyramidal neurons, performed 3D reconstructions (*Figure 7A,D*) and determined spine density, spine length and spine head diameter (see Materials and methods for details). Pyramidal neurons in acute (0 DIV) and cultured slices (2 DIV - 14 DIV) revealed no significant differences on average regarding spine density (acute and filled with biocytin: 0.54 ± 0.04 spines/µm, n = 5; cultured and filled with biocytin: 0.53 ± 0.04 spines/µm, p=0.99, n = 17; cultured and GFP-labeled: 0.44 ± 0.05, p=0.19, n = 11; Kruskis Wallis Test), spine length (acute biocytin: 1.46 ± 0.04 µm, n = 5; cultured biocytin: 1.54 ± 0.08 µm, n = 17, p=0.87; cultured GFP-labeled: 1.60 ± 0.05 µm, n = 11, p=0.45; Kruskis Wallis Test) or spine head diameter (acute biocytin: 0.47 ± 0.05 µm, n = 5; cultured biocytin: 0.52 ± 0.02 µm, p=052, n = 17; cultured GFP-labeled: 0.58 ± 0.04 µm, p=0.09, n = 11; Kruskis Wallis Test) (*Figure 7—figure supplement 1*). Since we did not find significant differences between any of the groups and particularly also not for biocytin versus GFP-driven analyses, we pooled GFP- labeled and biocytin filled neurons for linear regression analysis. We found a moderate negative correlation between the spine density and the DIV (*Figure 7E*, linear regression, $r^2 = -0.21$, p<0.01) and a moderate positive correlation between the spine head diameter and the DIV (*Figure 7E*, linear regression, $r^2 = 0.15$, p<0.05), while the average spine length showed no correlation to the DIV (*Figure 7E*, linear regression, p=0.99). In addition, we found a strong positive correlation between the distance of neuronal somata to the pia and the spine length (*Figure 7F*, linear regression, $r^2 = 0.46$, p<0.001), but not to the spine head diameter (*Figure 7F*, linear regression, p=0.06) or spine density (*Figure 7F*, linear regression, p=0.05).

In summary, analyses revealed mild changes at the most on spine level without clear indication of prevalent alterations related to the culturing process, suggesting that human brain slice cultures may represent a suitable model system to study pathophysiological mechanisms of neurological and neuropsychiatric disease.

## Virally transduced neurons maintain normal firing properties

To investigate whether AAV-transduction and GFP expression affected the electrophysiological properties of cortical neurons, we performed whole-cell patch clamp recordings of GFP positive neurons (n = 7) and GFP negative neurons (n = 13) at matching time points in culture (7–16 DIV) and assessed AP half width, resting membrane potential, sag potential and input resistance. All recorded neurons displayed normal AP firing upon supra-threshold current injections with either regular spiking patterns as typical for pyramidal neurons (*Figure 8C*, n = 7) or fast/adaptive spiking patterns as typical for interneurons (*Figure 8E*, n = 6) (*Ascoli et al., 2008*). Most of the parameters that we investigated in GFP positive neurons did not differ significantly from the values obtained from GFP negative control neurons (*Figure 8D and F*). We found however in the group of interneurons a significant reduction of the sag potential at hyperpolarization compared to GFP negative cells (*Figure 8F*), while all other parameters were not significantly different and while there were no significant differences found for any of the parameters in the pyramidal cell group. Taking together these results, virus transduction did not dramatically interfere with neuronal functionality.

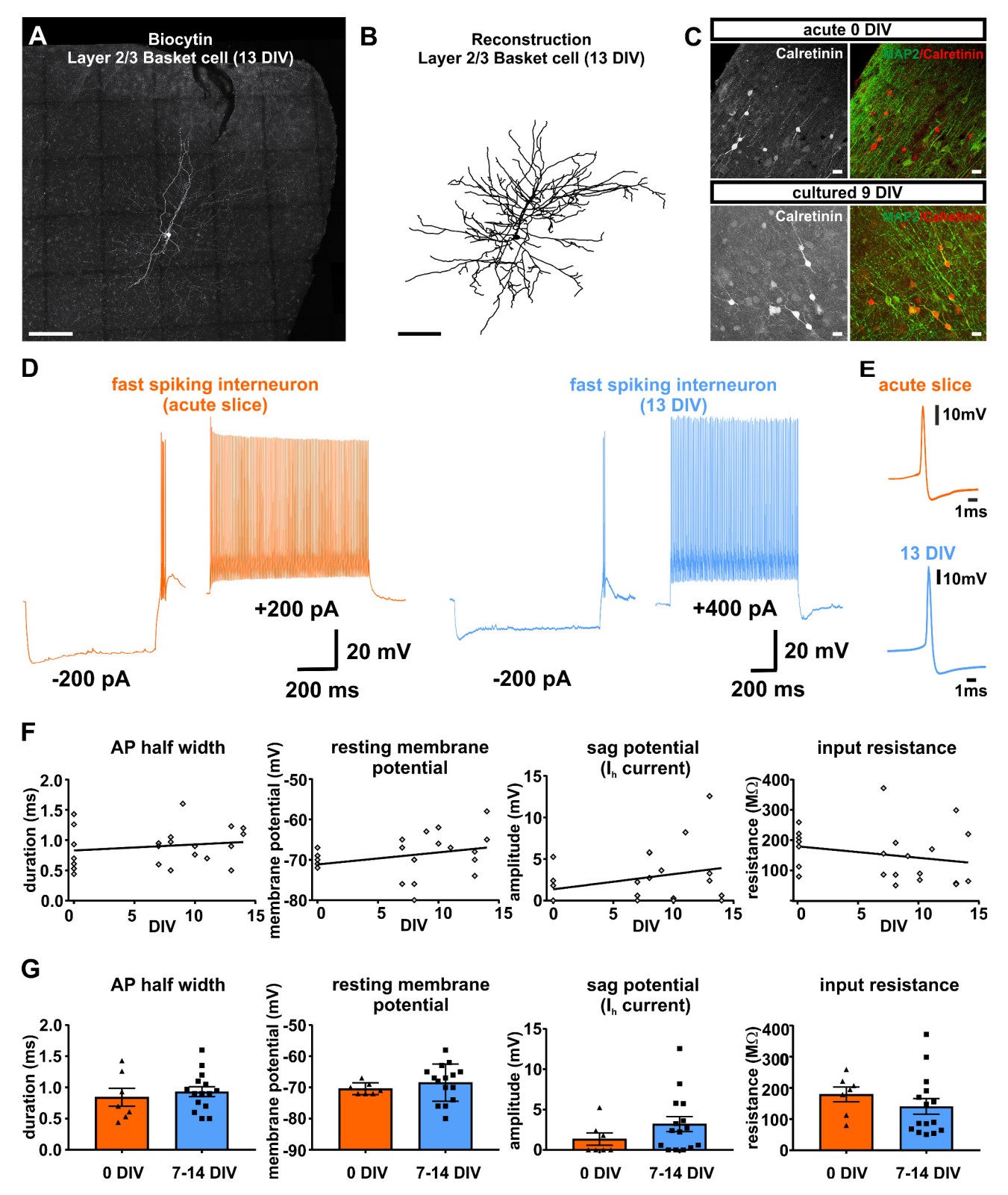

**Figure 6.** Presence and functionality of interneurons in human slice cultures. (**A**) Example of Layers 2/3 basket cell labled with biocytin and (**B**) after reconstruction, both scale bars 200 µm. (**C**) Staining of calretinin revealed presence of a subpopulation of inhibitory interneurons in acute (0 DIV) and late in culture (9 DIV), scale bar 20 µm. (**D**) Example of fast spiking interneuron firing in acute slices (orange) and late (blue) in culture (13 DIV). (**E**) Examples of interneuron APs in acute slice (orange) and late in culture (13 DIV, blue) reveal comparable AP half width. (**F**) Quantification and plotting of

*Figure 6 continued on next page*

*Figure 6 continued*

basic properties of IN in relation to the DIV. (**G**) Group comparison between the properties in acute slice and late in culture measured interneurons revealed no significant differences (Mann-Whitney test, p>0.05).

DOI: https://doi.org/10.7554/eLife.48417.009

## Two-photon live cell imaging of human brain slice cultures

In a last step we investigated whether virus-mediated GFP expression enables two-photon live cell imaging of adult human neurons within slice cultures. Slices were transferred into a submerged aCSF chamber and large tiles scans composed of detailed high-resolution stacks were acquired with a two-photon laser microscope (water-immersion x20 Objective, Laser tuned to 920 nm). In all examined slices (n = 5, 7–15 DIV) we could successfully identify pyramidal neurons (*Figure 9A* displays a representative example) including dendritic structures and emanating spines (*Figure 9A1*). In a second step we asked whether our system would be stable enough to enable time-lapse tracing of dendrites and spines of human pyramidal neurons over a time window of several hours. In two experiments (at 7 DIV and at 14 DIV) we imaged five distinct areas per slice for 24 hr and acquired image stacks every 30 min. While in three areas a slight shift in the z-axis resulted in an incomplete scan of the neurons and dendrites, further technical optimization allowed seven areas to be scanned successfully over the complete time course. *Figure 9* shows the 24 hours timepoint of the same pyramidal neuron displayed in *Figure 9A* at 0 hours of live cell imaging. *Figure 9B* and *Figure 9B1* show the same pyramidal neuron displayed in *Figure 9A* and *Figure 9A1* at 0 hours, now after 24 hours of live cell imaging. To further assess the stability of the system, dendrites of four imaged neurons were manually traced and reconstructed (*Figure 9—figure supplement 1A*) and total dendritic length and maximal sholl radius were semi-automatically determined for each of the four neurons at t = 0 hours and t = 24 hours. Data revealed reasonably small variability of parameters (mean ± SD: total dendritic length: −3.15 ± 10.87%; maximal sholl radius: 3.12 ± 6.50%) indicating feasibility of two-photon 24 hr life-cell imaging of human cortical slice cultures (*Figure 9—figure supplement 1B*).

After further optimization this approach will enable future studies of spine turnover of individual human neurons within brain slice cultures.

## Discussion

The human cerebral cortex is composed of many distinct cell types organized in very specific arrangements according to the highly specialized function of individual cells. Thus, questions are raised whether basic anatomic and physiological knowledge as well as disease related pathophysiological data obtained in animal models can be directly translated to the human CNS. For a long time, the ability to study bona fide human cortical neurons and investigate physiologically and pathologically relevant questions was confined to post mortem studies (*Anton-Sanchez et al., 2017*; *Elston and Fujita, 2014*; *Huttenlocher et al., 1982*) and to acute/fresh post-surgery tissue (*Beck et al., 2000*; *Marcuccilli et al., 2010*; *Verhoog et al., 2013*). With standard procedures, limited viability of resected tissue restricted electrophysiological recordings to a time window of 12–72 hr following surgery (*Wickham et al., 2018*). However, in recent years several methods have been incrementally established by us and other groups to keep human CNS tissue functionally intact for extended periods of days (*Ting et al., 2018*) to weeks (*Eugène et al., 2014*; *Andersson et al., 2016*; *Schwarz et al., 2017*). Initial studies based on development of special culturing strategies and complex artificial media revealed feasibility and prompted further investigation and optimization (*Eugène et al., 2014*). We recently demonstrated that hCSF significantly promotes neuronal viability and long-term survival of human brain slice cultures (*Schwarz et al., 2017*), allowing robust recordings of neuronal network activity for up to 21 DIV. These results indicated that after additional thorough characterization and further optimization such a system could potentially serve as a translational tool with broad application toward studies of human disease and development of novel therapeutic strategies. The success of such a model system would be predicted to not only critically depend on the viability of neurons but also on the maintenance of their distinct morphological characteristics. While we had demonstrated general neuronal survival within cultures (*Schwarz et al.,*

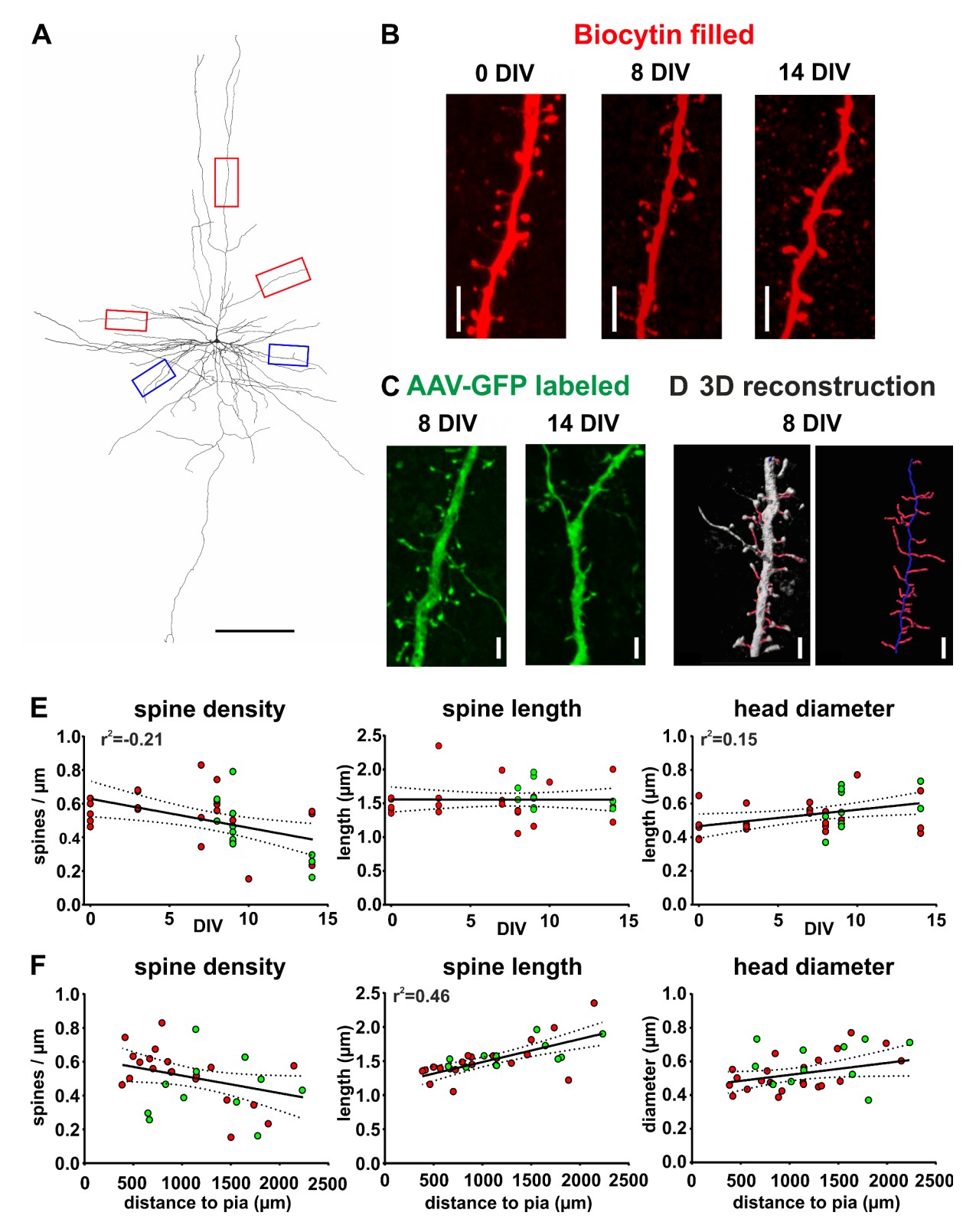

**Figure 7.** Spine measurements of human cortical pyramidal neurons in acute and cultured slices filled with biocytin and labeled with GFP. (A) 3D reconstruction of a typical layers 2/3 pyramidal neuron (biocytin filled, 7 DIV) in human brain slice cultures, scale bar 300 μm. Of each neuron five dendritic regions were chosen (three of the apical dendritic compartment, red boxes, and two of the basal dendritic tree, blue boxes). (B) Typical examples of spines localized on apical dendrites of representative layers 2/3 pyramidal neurons recorded and biocytin filled at 0 DIV (acute slice), at 8

*Figure 7 continued on next page*

*Figure 7 continued*

DIV and at 14 DIV, scale bar 5 µm. (C) Typical examples of spines of GFP-labeled neurons at 8 and 14 DIV, scale bar 5 µm. (D) For quantitative assessment, the z-stacks were 3D reconstructed and analyzed using NeuronStudio and Imaris software, scale bar 5 µm (see Materials and methods for details). (E, F) Neurons that were patched and filled with biocytin are represented by red circles and transduced GFP positive neurons by green circles. (E) Data reveal a moderate negative correlation of spine density with DIV and a moderate positive correlation of spine head diameter with DIV. (F) Plots of the same parameters versus the distance of the somata of the analyzed neurons to the pia revealed a strong positive correlation for spine length, but not the spine density or the spine head diameter.

DOI: https://doi.org/10.7554/eLife.48417.010

The following figure supplement is available for figure 7:

**Figure supplement 1.** Analysis of mean values of spine density, length and head diameter of biocytin filled and virally transduced GFP positive pyramidal neurons.

DOI: https://doi.org/10.7554/eLife.48417.011

*2017*) it remained elusive whether overall neuronal morphology, specifically dendritic complexity, spine distribution and designated structural features of spines, are preserved throughout the course of culturing. In the present study we now provide a systematic analysis of pyramidal neurons and interneurons across all cortical layers supporting overall electrophysiological and structural stability throughout the time window up to two weeks in culture. Specifically, we found that most parameters quantified in this study did not differ between neurons recorded in acute (0 DIV), early (2–3 DIV) or late stage cultures (7-14 DIV). One exception was the resting membrane potential of pyramidal neurons which was found to be slightly, but significantly lower for pyramidal neurons recorded in slices at 0 DIV. This might be interpreted as a small initial adaptation of pyramidal neurons to the new conditions in culture, particularly since we did not find continued depolarization of resting membrane potential in late stage cultures compared to early stage ones. Interestingly, interneurons did not seem to show this effect. Moreover, we measured and compared the properties of GFP positive neurons after viral transduction and found no major impairment of neuronal function in our analyzed samples of pyramidal neurons (n = 7) and interneurons (n = 6), which is in line with a previous study investigating virus transduction in human neurons (*Ting et al., 2018*). Although our data indicate no major interference of viral transduction with neuronal function we cannot rule out subtler alterations, particularly given the immense diversity of cortical neurons, which certainly cannot be represented in its entirety in our samples. This question needs to be further approached in future large-scale studies and respective investigations will benefit from community-based efforts. It has been reported before that pyramidal neurons of the human temporal cortex show a layer specific dendritic pattern with a gradual increase of the total dendritic length (both of the apical and basal compartment) of layers 2/3 pyramidal neurons correlating with the distance of the soma to the pia (*Mohan et al., 2015*). We confirmed this finding in our acute/fresh tissue samples and further demonstrate its robustness throughout the culturing period. Furthermore, we confirmed a decline in total apical dendritic length in our cultures at the intersection of layers 3 and 4, as demonstrated by *Mohan et al. (2015)*, corresponding to the presence and survival of spiny stellate neurons which are lacking an extended apical dendrite. While the gross morphology of the apical and basal dendritic arbors of pyramidal neurons are considered fairly stable over time *in vivo* and *in vitro* in acute slices, there is strong evidence that the dendritic fine structure undergoes a constant plastic remodeling (*Trachtenberg et al., 2002*; *Yuste and Bonhoeffer, 2004*). Especially the spines have been proven to be plastic and to appear and disappear on time scales of days in the developing (*Holtmaat and Svoboda, 2009*; *Lendvai et al., 2000*) and adult brain (*Trachtenberg et al., 2002*). On the other hand, despite this sustained turnover overall spine density has been shown to remain generally stable in adult cortex of mice (*Trachtenberg et al., 2002*). To investigate whether anatomic features of spines change over time in human slice cultures we performed high-resolution confocal imaging of dendritic compartments and assessed spine density, spine length and spine head diameter in correlation to the DIV. We detected only small alterations of the morphological spine parameters of neurons in cultured slices compared to neurons in acute slices along with only a moderate reduction of spine density. Although this is in line with the *in vivo* data by *Trachtenberg et al. (2002)*, this result may still seem surprising since cultured cortical tissue, disconnected from their surrounding circuitry and thus deprived from outside input, would rather be expected to significantly lose spines. However, these data are in line with *Trachtenberg et al. (2002)*, showing plastic remodeling in mouse pyramidal

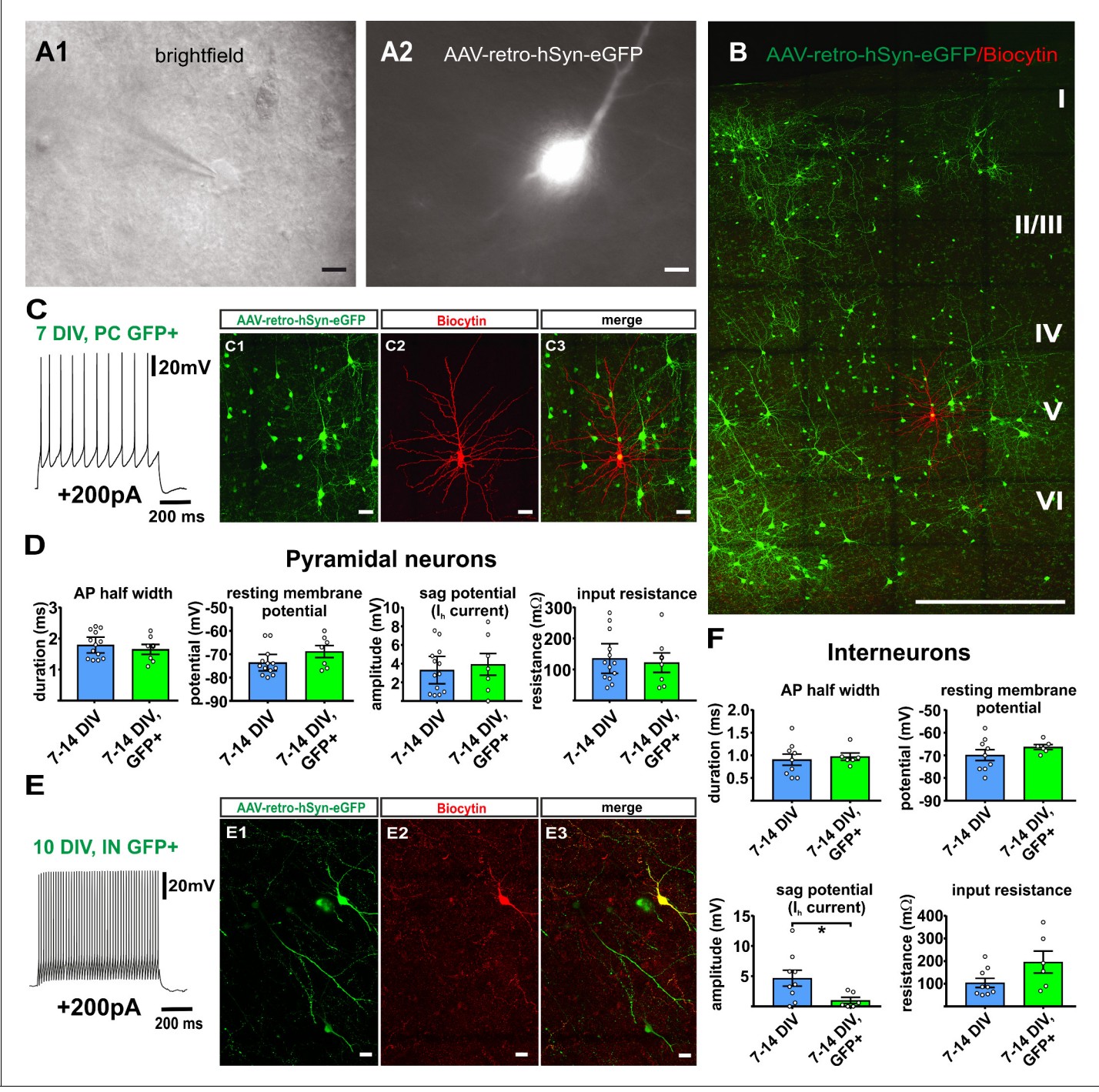

**Figure 8.** Virally transduced neurons show no major changes of basic properties. (**A**) GFP positive neurons were visualized according to their GFP fluorescence (**A2**) and targeted for whole-cell patch clamp recordings (**A1**), scale bars 10μm. (**B**) Overview image of a double positive pyramidal neuron (GFP and Biocytin) after whole cell recording and post-hoc staining, scale bar 500 μm; (**C1–C3**) and in detail, scale bar 20 μm. (**C**) Example of a representative regular firing pattern of a GFP positive pyramidal neuron (+200 pA). (**D**) Quantification of basic properties of GFP negative and GFP positive human pyramidal neurons. (**E**) Example of a GFP positive fast spiking interneuron. (**E1-E3**) GFP signal, biocytin filling and merge of the same neuron, scale bar 20 μm. (**F**) Quantification of basic properties of GFP negative and GFP positive human interneurons, *p<0.05, Mann Whitney test.
DOI: https://doi.org/10.7554/eLife.48417.012

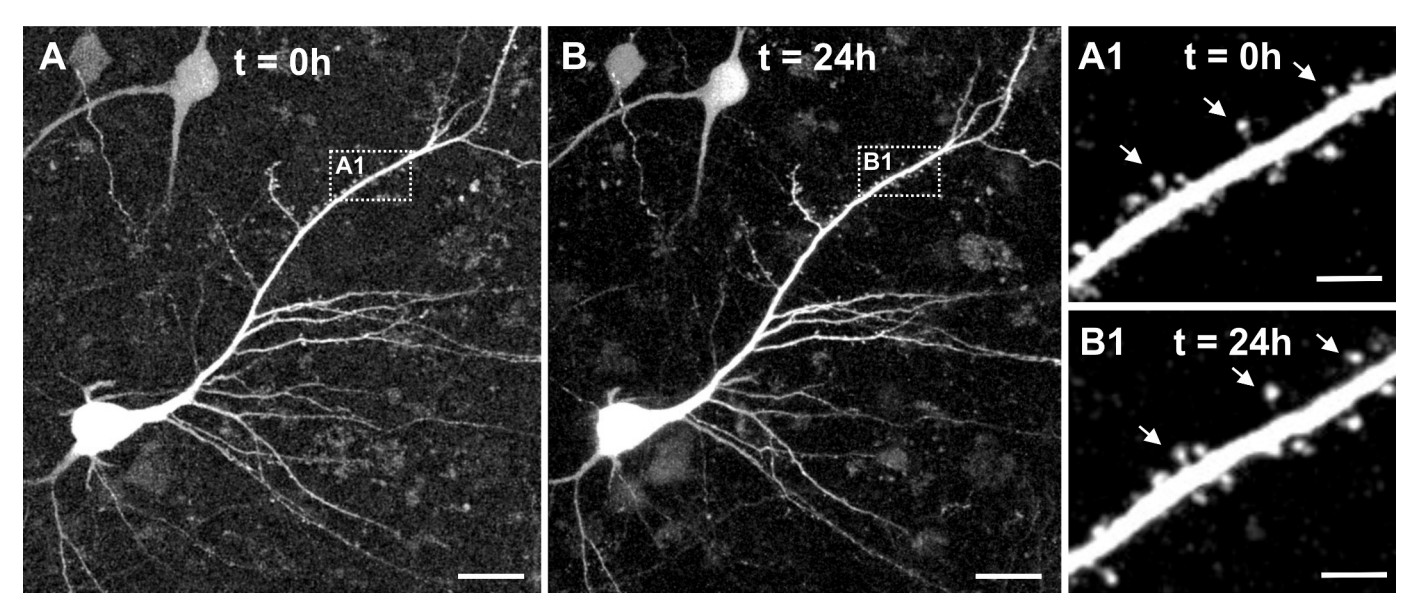

**Figure 9.** Two-photon live imaging of human neurons over 24 hr. (A) Maximum projection of a GFP positive human pyramidal neuron in slice culture at the start of the measurement and (B) after 24 hr life cell imaging with acquisition of image stacks every 30 min. (A1) and (B1) inserts are zoomed in images of the same branch of the apical dendrite at 0 and 24 hr of imaging. Arrows point to the same spines. Scale bars 20 μm (A, B) and 5 μm (A1, B1).

DOI: https://doi.org/10.7554/eLife.48417.013

The following figure supplement is available for figure 9:

**Figure supplement 1.** Filament measurements reveal stability over the time of 24 hr.
DOI: https://doi.org/10.7554/eLife.48417.014

neurons with an increased spine turnover rate and number of transient spines but overall stability of spine density in response to activity deprivation. Such a homeostatic mechanism could explain the relative stability of spine density in adult human brain slice cultures reported here. Future experiments directly determining spine turnover by two-photon live cell imaging will be needed to corroborate this hypothesis and to investigate whether alternative or additional mechanisms could have a role in maintenance of spine density in human pyramidal neurons (in slice cultures). Such studies are indeed conceivable and we provide first proof-of-concept data regarding feasibility of live cell imaging for extended time periods (up to 24 hr) of human brain slice cultures by taking advantage of virally mediated neuronal GFP expression. Furthermore, we demonstrate genetic labeling as an efficient approach toward 3D reconstruction and detailed morphological analysis of relatively large quantities of adult human neurons per tissue sample, providing a valid alternative to laborious single cell biocytin fillings. Post processing of the tissue for GFP signal amplification by standard immunocytochemistry or in combination with tyramide boosting will likely provide structural information even surpassing the one available from classic biocytin fillings. Depending on the degree of cell type specificity of the promoters used to drive the expression of fluorescent proteins this strategy may also enable screening for rare human CNS cell types or targeting of certain subpopulations like interneurons (*Dimidschstein et al., 2016*). It seems very intriguing to adopt the same set of tools to study cell autonomous and (in light of substantial viral transduction rates) potentially even cell non-autonomous effects of disease related mutated proteins on human CNS cell types of interest. Adapting state-of-the-art molecular genetic tools (such as optogenetic and chemogenetic approaches or homologous recombination-driven genetic targeting), currently available in model organisms, for application in human CNS tissue appears increasingly practical and would open completely new opportunities to study both physiological properties of human CNS circuitry and pathological conditions underlying neurological disease.

In summary, we found a remarkable stability of morphological and electrophysiological parameters of human pyramidal neurons as cortical network components of slice cultures over a time course of 14 DIV. Furthermore, we demonstrate widespread viral transduction of human brain slice cultures, enabling genetic manipulation of adult human neurons under very controlled conditions. We propose that this model system will be of great value for many investigations in viable human neuronal networks such as therapeutic screening, investigation of electrophysiological and structural properties of human CNS circuitry and direct exploration of mechanisms underlying neurological disease.

## Materials and methods

### Patients

Human neocortical brain slice cultures were prepared from access tissue (cortical tissue outside the epileptic focus, resected in order to gain access to the pathology) obtained from patients undergoing epilepsy surgery. For this study, we collected and included data of 15 patients (*Table 1*). All patients were surgically treated for intractable epilepsy, in two patients the histology revealed low-grade tumors (Ganglioglioma WHO Grade I; Oligodendroglioma, WHO Grade II).

### Preparation of slices from resected tissue

Approval (# 338/2016A) of the ethics committee of the University of Tübingen as well as written informed consent was obtained from all patients, allowing spare tissue from resective surgery to be included in the study. Tissue preparation was performed according to published protocols (*Verhoog et al., 2013*). Cortex was carefully microdissected and resected with only minimal use of bipolar forceps to ensure tissue integrity, directly transferred into ice-cold artificial cerebrospinal fluid (aCSF) (in mM: 110 choline chloride, 26 NaHCO$_3$, 10 D-glucose, 11.6 Na-ascorbate, 7 MgCl$_2$, 3.1 Na-pyruvate, 2.5 KCl, 1.25 NaH$_2$PO$_4$, und 0.5 CaCl$_2$) equilibrated with carbogen (95% O$_2$, 5% CO$_2$) and immediately transported to the laboratory. Tissue was kept submerged in cool and carbogenated aCSF at all times. After removal of the pia, tissue chunks were trimmed perpendicular to the cortical surface and 250–350 µm thick acute slices were prepared using a Microm HM 650V vibratome (Thermo Fisher Scientific Inc) (*Figure 1—figure supplement 2A*). Afterwards the slices were kept in aCSF equilibrated with carbogen for 0.5 hr at room temperature before they were

**Table 1.** Tissue samples included in this study.

| Age at surgery | Gender | Resected brain area | Number and type of cells included in this study |
|---|---|---|---|
| 30 | M | temporal lobe | 3 PC |
| 30 | M | temporal lobe | 2 PC |
| 10 | F | temporal lobe | 3 PC + 3 IN |
| 38 | F | temporal lobe | 2 IN |
| 16 | M | temporal lobe | 1 PC |
| 18 | M | frontal lobe | 6 PC |
| 40 | F | temporal lobe | 4 PC + 3 IN |
| 40 | F | frontal lobe | 1 IN |
| 29 | M | temporal lobe | 5 PC |
| 56 | F | temporal lobe | 3 PC |
| 31 | M | temporal lobe | 3 PC + 2 IN |
| 44 | M | temporal lobe | 5 PC + 3 IN |
| 46 | F | temporal lobe | 2 PC + 1 IN |
| 67 | M | temporal lobe | 2 PC + 1 IN |
| 57 | M | frontal lobe | 6 PC + 6 IN |

DOI: https://doi.org/10.7554/eLife.48417.015

transferred either to patch setups for acute electrophysiological recordings and biocytin fillings or onto culture membranes for cultivation.

## Human brain slice cultures

After the cortical tissue was sliced as described above, slices were cut into several evenly sized pieces (~1.0×1.0 cm, *Figure 1—figure supplement 2A*). Subsequently, slices were transferred onto culture membranes (uncoated 30 mm Millicell-CM tissue culture inserts with 0.4 µm pores, Millipore) and kept in sixwell culture dishes (BD Biosciences). For the first hour following the slicing procedure the slices were cultured in 1.5 ml NSC media (48% DMEM/F-12 (Life Technologies), 48% Neurobasal (Life Technologies), 1x N-2 (Capricorn Scientific), 1x B-27 (Capricorn Scientific), 1x Glutamax (Life Technologies), 1x NEAA (Life Technologies) + 20 mM HEPES before changing to 1.5 ml hCSF per well without any supplements. No antibiotics or antymicotics were used during cultivation. The plates were stored in an incubator (ThermoScientific) at 37°C, 5% $CO_2$ and 100% humidity. For electrophysiological recordings slice cultures were transferred into the recording chamber of a patch clamp rig. After finishing patch clamp recordings, slices were fixed in 4% paraformaldehyde (PFA) and processed for immunocytochemistry (ICC), as described in detail in the ICC section.

## Structural features of acute cortical slices and slice cultures

*Mohan et al. (2015)* and subsequently *Ting et al. (2018)* and *Goriounova et al. (2018)* provided a framework defining cortical layering and layer boundaries of human cortical brain slices based on distance from the pia. This system can be used to register neurons according to the distance of their soma position from the pial surface, thereby assigning neurons to cortical layers 2/3, 4 or 5/6. To confirm comparability to our acute (0 DIV) tissue and in order to assess whether this system can be also directly applied to our tissue cultures, we compared the thickness of cortical gray matter between tissue slices originating from independent surgeries and between different time points in culture. Following DAPI staining of slices (0 DIV - 15 DIV) layer four was clearly visible as delineated band (as has previously been demonstrated by *Ting et al., 2018*)~1200 µm below the pial surface (*Figure 1—figure supplement 2E*). The distance of the outer margin (edge) of layer four from the pia was assessed for tissue slices prepared from three independent surgeries (P1-3). Mean values were plotted as bar diagrams for comparison (*Figure 1—figure supplement 2B*) and did not reveal significant differences. Furthermore, this distance did not change significantly over time in culture up to the endpoint of analysis at 20 DIV (*Figure 1—figure supplement 2C*). Importantly, results are in line with the published data referenced above. While mouse organotypic slice cultures are known to flatten out quite extensively throughout the time in culture to the point that they may even become translucent, human brain slice cultures revealed only a mild decrease of thickness of about 20% over the analyzed three-week period in culture (*Figure 1—figure supplement 2D*).

Assessment of human brain slice thickness: Slices were fixed with 4% PFA and thickness was determined by two-photon microscopy based on detection of either GFP-signal or tissue autofluorescence. Imaging was performed on an upright Zeiss LSM 880 NLO microscope, using an IR-optimized water-immersion x20 Objective (W Plan-Apochromat x20/1.0, Carl Zeiss, Jena). A MaiTai eHP DeepSee (Spectra Physics) laser was used for two-photon excitation, tuned at 920 nm for exciting GFP, YFP, hFTAA and Atto550 fluorescence. Emission was collected using Non-Descanned GaAsP detectors, separated by a 520/50 emission filter. Images were collected at a resolution of 0.208 µm x 0.208 µm x 0.640 µm per voxel. Two z-stacks at different positions of at least two slices per DIV time point were acquired and analyzed with Imaris to determine the thickness of the 3D-reconstructed image. To exclude any effects of the fixation procedure on slice thickness, late stage slice cultures (20 DIV) were first imaged unfixed via two-photon live-cell imaging and then re-imaged after fixation with 4% PFA.

## Classification of neurons according to cortical layering

*Mohan et al. (2015)*, neurons were classified by the distance of their soma to the pia as layers 2/3 (distance to pia: 300–1200 µm), layer 4 (distance to pia: 1200–1500 µm) or layers 5/6 (distance to pia: 1500–2900 µm) pyramidal neurons.

## Collection of human cerebrospinal fluid (hCSF)

hCSF was collected from patients with normal pressure hydrocephalus (NPH) or idiopathic intracranial hypertension. We received and pooled hCSF of several patients who needed to undergo a CSF tap test either by lumbar puncture or lumbar drain as part of the diagnostic or therapeutic workup. Approval of the ethics committee of the University of Tübingen as well as written informed consent from all patients was obtained. It is well established and known from daily clinical practice that hCSF of NPH patients exhibits physiological/normal hCSF parameters (lactate, glucose, cell count, protein levels – see *Schwarz et al., 2017*), undistinguishable from the ones of healthy individuals (*Bjorefeldt et al., 2015*). The hCSF was centrifuged at 4000 rpm at 4°C for 10 min. Samples were sterile filtered using a 0.2 µm syringe filter and stored at −80°C.

## Whole-cell patch clamp recordings

Slices were positioned in a submerged-type recording chamber (Scientifica, United Kingdom/Warner apparatus), continuously superfused with 32.0 + /- 1.5°C recording aCSF (in mM: 118 NaCl, 25 NaHCO$_3$, 30 D-glucose, 1 MgCl$_2$, 3 KCl, 1 NaH$_2$PO$_4$, und 1.5 CaCl$_2$) and visualized with a BX61WI Microscope (Olympus) or a Leica stereomicroscope. After 1 hr of equilibration in the patch setup recordings were performed using recording electrodes with a resistance of 3–5 MΩ and filled with an intracellular whole-cell patch-clamp pipette solution containing the following components (in mM): 140 K-gluconic acid, 1 CaCl$_2$*6H$_2$O, 10 EGTA, 2 MgCl$_2$*6H$_2$O, 4 Na$_2$ATP, and 10 HEPES, pH 7.2, 300 mOsm. The intracellular whole-cell patch-clamp pipette solution contained biocytin (5 mg/ml, Sigma, B4261) to allow for post-hoc identification of the location and morphology of recorded neurons. Resting membrane potential was recorded immediately after breaking the seal and half-witdh of the action potential was measured 5 min after breaking the seal. Whole-cell current-clamp recordings were obtained from cortical neurons using either the visual-patch or blind patch technique and sampled at 20–100 kHz with a low-pass filter of 5–30 kHz. Recordings were performed with unpolished patch electrodes manufactured from borosilicate glass pipettes with filament (Science products). Patch-clamp experiments were performed with a patch-clamp amplifier (Multiclamp 200B) or a NPI Bridge Amplifier (Model BA-01X), a digitizing interface (Digidata 1440A or 1550A Digidata), and pClamp 10 software (Molecular Devices). The junction potential was calculated and subtracted offline to correct the membrane potential in current clamp mode. After recording, the slices were placed in neutral buffered 4% PFA solution at 4°C overnight for fixation followed by three rinses in phosphate buffered saline (PBS) and by subsequent immunocytochemical staining procedures. The patched neurons were filled with the intracellular solution for a minimum of 10 min during the electrophysiological recordings allowing full morphological reconstruction. Subsequently, the fixed slices were exposed to streptavidin-Cy3 (1:100 of 1 mg/ml, Sigma, S6402) for 3 hr at room temperature and mounted on glass coverslips for visualization of the biocytin filled neurons (*Figure 1A*).

## Viral transduction

AAV-retro(AAVrg)-hSynapsin-GFP-virus was purchased from Addgene (pAAV-hSyn-GFP was a gift from Bryan Roth - Addgene viral prep # 50465-AAVrg) and had a titer of 10$^{12}$. The virus was injected in multiple areas of the slice using a picospritzer (PDES-O2DX/NPI electronics, Tamm, Germany) at 3–5 days *in vitro* (DIV).

## 3D-Reconstruction of neurons

To reconstruct the full morphology of biocytin filled (*Figure 1A*) or GFP-labeled neurons (*Figure 2*), imaging was performed on a Leica TCS SP8 confocal laser-scanning microscope (Bensheim, Germany) using a 40x/1.3NA oil-immersion objective. Z-stack tile scans of the whole culture slice were acquired. The Cy3 fluorescence was excited at 548 nm using a tunable white light laser and emission was detected between 555 nm and 615 nm. The GFP signal was excited at 488 nm and emission was detected between 493 nm and 530 nm. For a precise three-dimensional reconstruction of the dendritic segments, the slices were imaged at a voxel size of 0.2481 × 0.2481×0.7 µm. The tile scans were stitched using ImageJ and the plugin grid-stitching (*Preibisch et al., 2009*). Filament reconstruction was performed using Imaris x64 9.0.2 software (RRID:SCR_007370), enabling precise quantification of dendritic length. The number of dendrites radiating from the soma of one neuron was

thoroughly checked for misalignment and false connections using the rotating 3D fluorescence intensity image in Imaris. The apical and the basal dendrites as well as the axons were identified manually and labeled, represented in the colors red (apical dendrites), blue (basal dendrites) and magenta (axons) and then quantified separately for total length.

## Spine morphology

To determine the spine morphology of the neurons we performed high-resolution scans of five selected dendritic areas of the cells. We always chose three areas of the apical dendrites (*Figure 7A*, red boxes, red boxes) and two of the basal dendrites (*Figure 7A*, blue boxes, blue boxes). The voxel size for all analyzed stacks was $0.071 \times 0.071 \times 0.1998$ µm and a z-stack of 10–15 µm was taken. These stacks were further analyzed using NeuronStudio software (*Rodriguez et al., 2008*). Of each analyzed area the spines were detected using the NeuronStudio algorithm (with a minimal cutoff of 0.4 µm and maximum cutoff of 6 µm for the spine size) and subsequently manually corrected if necessary. We calculated the spine density (total number of spines/length of the analyzed dendrite), the mean value of spine length and spine head diameter of each of the five selected dendritic areas. Values of all 5 areas per neuron were averaged and plotted against the "DIV" or against the "distance to pia" such that every data point in *Figure 7E* and *Figure 7F* corresponds to the average of the 5 areas per neuron. We have chosen this method to minimize potential measuring errors for each analyzed neuron.

## Immunocytochemistry

For ICC, slices were incubated for 20 min in 50% methanol with 1% $H_2O_2$ to quench endogenous peroxidases. The slices were rinsed with PBS supplemented with 0.2% Triton X-100 (T-PBS) and blocked for 1 hr in T-PBS, 0.4 g merthiolate and 1% normal goat serum before adding primary antibodies in respective dilutions for 2–3 days at 4°C. Slices were rinsed three times in T-PBS and then incubated in Alexa-fluorophore-conjugated secondary antibodies in T-PBS, 0.4 g merthiolate and 1% normal goat serum overnight at 4°C. Slices were rinsed three times with T-PBS before staining with DAPI (1:5000 in PBS for 2–5 min). After four final rinses (3x PBS, 1x Ampuwa water), slices were mounted in Fluoromount G (SouthernBiotech) on glass slides. Primary and secondary antibodies were used at the following dilutions: mouse anti-Satb2 for excitatory neurons (1:100, Abcam, ab51502, RRID:AB_882455), chicken anti-MAP2 for the somatodendritic compartment of neurons (1:500, Abcam, ab5392, RRID:AB_2138153); chicken anti-Calretinin (1:500, Merck, AB1550, RRID: AB_90764); goat anti-mouse IgG Alexa 568 (1:500, Thermo Fisher, A-11004, RRID:AB_2534072); goat anti-chicken Alexa 488 (1:500, Thermo Fisher, A-11039, RRID:AB_142924) and donkey anti-goat Alexa 568 (1:500, Thermo Fisher, A-11057, RRID:AB_142581).

## Cell counts and quantification

Images for Satb2 and Map2 quantification were taken with a Leica TCS SP8 confocal laser-scanning microscope (Bensheim, Germany) using a 40x/1.3NA oil-immersion objective. We obtained images of four to six areas in layers 2/3 (each area 290 µm x 290 µm in size) per slice and analyzed maximum projections of 2.1 µm stacks. The absolute number of cells per area was normalized to 100 µm X 100 µm (10000 µm²) and plotted. The quantification was performed for the same dataset in a blinded manner by three independent researchers.

## Two-photon live cell imaging and sholl analysis

Slice cultures with GFP-positive neurons were imaged on an upright Zeiss LSM 880 NLO microscope, using an IR-optimized water-immersion x20 Objective (W Plan-Apochromat x20/1.0, Carl Zeiss, Jena). A MaiTai eHP (Spectra Physics) laser was used for two-photon excitation, tuned at 920 nm for exciting the GFP fluorescence. Emission was collected using the Non-Descanned GaAsP detection path, with a 520/50 emission filter. A 3D tile scan of a larger area of the culture was setup in the ZEN black software (Carl Zeiss, Jena), collecting images at a resolution of $0.415 \times 0.415 \times 1.2$ µm per voxel. The above configuration was optimized for highest light throughput, ensuring the minimum possible laser intensity, in order to avoid thermal damage to the neurons. For 24 hr time-lapse imaging stacks of images were acquired every 30 min at a resolution of $0.208 \times 0.208 \times 0.640$ µm per voxel. Five areas were marked in the experiment designer of the Zen black software so that the

system could navigate to all positions and image predefined areas in an automated manner. The experiment designer was setup to repeat this chain of acquisitions every 30 min.

To guarantee a stable environment while imaging, the culture inserts were placed in a custom-made chamber, designed to securely hold the insert at the same orientation and position, submerged in aCSF. Prewarmed, carbogenated aCSF constantly perfused the chamber. The customized stage-top incubator which housed the customized culture insert holder was kept at 37°C and was filled with humidified carbogen.

All movies acquired with two-photon live-cell imaging were deconvolved with Huygens Essential (Scientific Volume Imaging B.V., Hilversum) before further processing. After deconvolution, images and movies were analyzed using Imaris (Bitplane, Belfast, RRID:SCR_007370). The filaments tool of the Imaris software was used to semi-automatically trace the dendrites and spines of the neurons. The total length of the filaments was automatically calculated in Imaris. For the sholl analysis of the traced dendrites in Imaris, the sholl radius was set to 2 µm. The maximum radius of the sholl analysis was defined as the maximum radius in the field of view at which an intersection of the filament with one of the sholl circles could still be detected.

## Analysis and statistics

Statistical analyses were performed with Graph Pad Prism seven using paired or unpaired Mann Whitney test to compare two groups and Kruskal-Wallis test with Dunn's Multiple comparisons test for three groups. Mean values ± standard error of the mean (SEM) are presented in text and Figures, unless specified differently. Linear regressions were analyzed for the correlations between properties and the days *in vitro* and the distance to the pia (DTP).

## Acknowledgements

The authors thank all patients for participating in this study. We thank Evelyn Dubois, Elke Stransky and Christina Tomschitz for their assistance with collecting hCSF samples. The study was supported by the German Research Foundation (KO-4877/2–1, WE4896/3-1 and WE4896/4-1; FOR 2715) and by the ministry of rural affairs and consumer protection (award to NS supporting the replacement, refinement and reduction of animal experiments). NS was in part supported by an intramural research funding program of the Faculty of Medicine, University of Tübingen (Fortüne 2381-0-0). TVW was supported by an intramural Clinician Scientist fellowship granted by the Faculty of Medicine, University of Tübingen (419-0-0).

## Additional information

### Funding

| Funder | Grant reference number | Author |
|---|---|---|
| Eberhard Karls Universität Tübingen | Fortüne 2381-0-0 | Niklas Schwarz |
| German Research Foundation | KO-4877/2-1 | Henner Koch |
| German Research Foundation | WE4896/3-1 | Yvonne G Weber |
| Eberhard Karls Universität Tübingen | Clinician Scientist fellowship 419-0-0 | Thomas V Wuttke |
| German Research Foundation | FOR 2715 | Ulrike BS Hedrich Albert J Becker |
| German Research Foundation | WE4896/4-1 | Yvonne G Weber |

The funders had no role in study design, data collection and interpretation, or the decision to submit the work for publication.

### Author contributions

Niklas Schwarz, Henner Koch, Conceptualization, Data curation, Formal analysis, Funding acquisition, Investigation, Methodology, Project administration, Resources, Software, Supervision,

Validation, Visualization, Writing - original draft, Writing - review and editing; Betül Uysal, Jacqueline C Bahr, Formal analysis, Investigation; Marc Welzer, Nikolas Layer, Heidi Löffler, Kornelijus Stanaitis, Harshad PA, Investigation; Yvonne G Weber, Conceptualization, Funding acquisition, Resources; Ulrike BS Hedrich, Conceptualization, Investigation, Visualization, Writing - original draft; Jürgen B Honegger, Resources; Angelos Skodras, Investigation, Software; Albert J Becker, Writing - original draft; Thomas V Wuttke, Conceptualization, Funding acquisition, Methodology, Project administration, Resources, Supervision, Validation, Writing - original draft, Writing - review & editing

### Author ORCIDs
Niklas Schwarz https://orcid.org/0000-0002-4064-3073
Thomas V Wuttke https://orcid.org/0000-0001-5655-8490
Henner Koch https://orcid.org/0000-0002-6883-3071

### Ethics
Human subjects: Approval (# 338/2016A) of the ethics committee of the University of Tübingen as well as written informed consent was obtained from all patients, allowing spare tissue from resective surgery to be included in the study.

### Decision letter and Author response
Decision letter https://doi.org/10.7554/eLife.48417.020
Author response https://doi.org/10.7554/eLife.48417.021

## Additional files
### Supplementary files
• Transparent reporting form
DOI: https://doi.org/10.7554/eLife.48417.016

### Data availability
Reconstructions of GFP labeled and Biocytin filled neurons and spine details of the analyzed cells have been deposited in: https://doi.org/10.5061/dryad.s5g2712.

The following dataset was generated:

| Author(s) | Year | Dataset title | Dataset URL | Database and Identifier |
|---|---|---|---|---|
| Niklas Schwarz, Betül Uysal, Marc Welzer, Jacqueline Bahr, Nikolas Layer, Heidi Löffler, Kornelijus Stanaitis, Harshad PA, Yvonne G Weber, Ulrike BS Hedrich, Jürgen B Honegger, Angelos Skodras, Albert J Becker, Thomas V Wuttke, Henner Koch | 2019 | Reconstructions of GFP labeled and Biocytin filled neurons and spine details of the analyzed cells | https://doi.org/10.5061/dryad.s5g2712 | Dryad Digital Repository, 10.5061/dryad.s5g2712 |

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
