## [Decision Letter]

Thank you for submitting your article "Long-term adult human brain slice cultures – a model system to study human CNS circuitry and disease" for consideration by *eLife*. Your article has been reviewed by three peer reviewers, and the evaluation has been overseen by a Reviewing Editor and Eve Marder as the Senior Editor. The following individual involved in review of your submission has agreed to reveal their identity: Huibert D Mansvelder (Reviewer #3).

The reviewers have discussed the reviews with one another and the Reviewing Editor has drafted this decision to help you prepare a revised submission.

Summary:

This manuscript describes physiological and anatomical comparisons between acute human cortical slices and those kept in culture for up to 2 weeks. The authors use AAV-mediated expression of GFP to show no substantial changes in basic whole cell current clamp properties, large-scale dendritic anatomy, or fine scale synapse morphology in samples of neurons from the acute and cultured conditions. This article is a welcome contribution to an exciting field and validates the use of human resected tissue for functional long-term translational studies, as well as for basic scientific questions on how human neuronal networks are organized. While the reviewers appreciate the advance of being able to culture human slices for weeks, particularly when combined with AAV mediated transfection, they suggest to perform additional experiments to convincingly and rigorously demonstrate that long term culture does not substantially alter the physiology or morphology of the neurons over the long timescale. The results for the 14 DIV time-point came from very low cell number rather than appropriate and comprehensive population and statistical comparisons.

Essential revisions:

1) Much of the data seems phenomenological, rather than systematic and quantitative. For example, the authors state: "To investigate whether AAV-transduction and GFP expression affected the electrophysiological properties of cortical neurons, we performed whole-cell patch clamp recordings of GFP positive neurons. All recorded neurons displayed normal AP firing upon supra-threshold current injections with either regular spiking pattern (Figure 8B, n=3) or fast spiking pattern (n=3). In addition, synaptic inputs (spontaneous postsynaptic currents (sPSCs)) were present in voltage clamp mode in all GFP positive neurons (n=6) (Figure 8B). Taking together these results, virus transduction did not interfere with neuronal functionality." N = 3 neurons is an extremely low number, and the authors do not say how many slices or patients these 3 neurons came from. I also don't see any statistics for these comparisons, and no real population data in the figures. I do not think this is a rigorous test of the authors' contention that virus transduction does not interfere with neuronal functionality.

2) The spine analysis is confusing. It's not clear what is being analyzed – is each data point in Figure 7 an averaged value of spine measurements per neuron? Where were the spine measurements sampled from in the neurons? The authors need to improve the clarity regarding how these experiments were conducted.

3) The Materials and methods are incomplete and would not allow a reader to repeat the experiment. For example, there is no mention of what media the slices were cultured in. Or how the slices were kept sterile, which seems like a key point.

How were the slices treated after sectioning?

Were they recovered in a warm incubation system as is convention? If so, what temperatures and times were used?

Were the cultured slices sent directly to culture after slicing? After recovery? How were they treated? This are crucial points for this methods-oriented paper and the authors are extremely light on the details.

When were electrophysiological recordings (and biocytin filling) of acute slices done after cutting? From experience of both human and rodent slice recordings, slices have to rest after slicing to recover due to processes like spine retraction and glycogen depletion (e.g. Kirov et al., 2004, Neuroscience), usually an hour for rodents and 3 hours for human hippocampal and cortical tissue. This should be described in Materials and methods.

4) The authors refer to the choline chloride solution they use to transport and slice the human tissue as aCSF, but they also refer to their patch clamp recording solution (which they do not define) as aCSF. It's highly unlikely they performed physiology experiments in the choline solution. Describing the constituency of the recording solution is necessary.

5) Variation in patient material needs more attention.

Patient material can be quite variable depending on various factors, such as the level of pathophysiology of the tissue, patient age and handling during surgery. In this manuscript there are 3 different cortical areas measured, the age range is from 10 to 56 and 2 patients presented with gangliogliomas. This can be a problem when samples sizes are low if the slices are all from the same patients and not representing the majority. The patient material underlying the analyses should be stated (with patients numbered) when describing groups and figures.

6) The comparison between DIV 0 and DIV 2-14 for electrophysiology is a bit strange. If the goal is to investigate effect of time in culture should the comparison be between DIV 0/2 and DIV 14? The numbers of cells are not enough for this comparison. On the same note the time-point for MAP2/Satb2-stainings were 9 DIV vs 14 DIV. If changes in neuronal number have occurred before 9 DIV this will not be detected and MAP2/Satb2 data would not support the statement of a robust survival of neurons throughout the culturing time.

7) The study mainly focusses on pyramidal neurons. Although the Satb2/Map2 comparisons in Figure 1—figure supplement 2 do hint at stable presence of inhibitory cells in the culture, it would be a very valuable addition to include analysis on properties of inhibitory interneurons to the study. In the description of Figure 8, recordings from fast spiking neurons are mentioned, but not shown. Are morphological and/or physiological properties of interneurons equally well-maintained across the culturing period? Perhaps this can be addressed in the data the authors already have collected?

8) It is stated that patched neurons were filled with the intracellular solution for a minimum of 10 minutes "allowing for full morphological reconstruction". My question is if 10 minutes is sufficient for diffusion of biocytin to reach distal dendrites? What would be the diffusion rate of biocytin in your setup? This could be estimated using the recording temperature, which is missing in Materials and methods.

9) Related to Figure 1G: If brain slice properties alter over culture time, for instance connectivity or cellular morphology, this could show up in input resistance or membrane time constant. Were these parameters affected by days in culture? Please also test for potential correlations between input resistance and membrane time constant change with amount of time in culture.

10) From culturing rodent cortical brain slices it is known that slices flatten out during culture period. Is the three dimensional structure maintained during 3 weeks of adult human slice culture? Does slice thickness decrease over 3 weeks culturing? Please provide numbers.

11) In Figure 3 neurons were assigned to cortical layers based on depth from pia. Were layer dimensions stable over the entire culture period? This could be determined by Nissl stains at different DIVs. Please provide numbers.

12) Figure 4: reconstructing GFP positive neurons may be hampered by overlapping processes containing GFP from neighboring neurons. Can the authors rule out that 'contamination' of dendritic structure reconstructions by overlap of stained processes from different individual neurons affected their quantification?

13) In Figure 5C, right panel, the comparisons of the total dendritic length show a slight difference between the reconstructed biocytin and GFP labelled cells. The authors mention that 'very distal dendritic ramifications being captured less reliably'. It is not clear to me what would be the cause of this reduced reliability. Please explain.

14) In Figure 8: please provide quantitative support of the statements that physiological properties and AP firing of GFP positive neurons was 'normal'.

15) Were synaptic current frequencies and amplitudes comparable to GFP-negative neurons at different DIVs?

16) In the final paragraph of the Results section, the authors hint at possibility to follow spine morphology changes in time, possibly across days. For this it is required that the authors succeed in imaging the same stretch of dendrite across several days. It would be a nice addition to the study if the authors could provide data to support this possibility.

---

## [Author Response]

Essential revisions:1) Much of the data seems phenomenological, rather than systematic and quantitative. For example, the authors state: "To investigate whether AAV-transduction and GFP expression affected the electrophysiological properties of cortical neurons, we performed whole-cell patch clamp recordings of GFP positive neurons. All recorded neurons displayed normal AP firing upon supra-threshold current injections with either regular spiking pattern (Figure 8B, n=3) or fast spiking pattern (n=3). In addition, synaptic inputs (spontaneous postsynaptic currents (sPSCs)) were present in voltage clamp mode in all GFP positive neurons (n=6) (Figure 8B). Taking together these results, virus transduction did not interfere with neuronal functionality." N = 3 neurons is an extremely low number, and the authors do not say how many slices or patients these 3 neurons came from. I also don't see any statistics for these comparisons, and no real population data in the figures. I do not think this is a rigorous test of the authors' contention that virus transduction does not interfere with neuronal functionality.

We agree with the reviewer and increased the sample sizes. We doubled the number of analyzed pyramidal neurons after virus transduction from n=3 to n=7. Non-transduced GFP negative (n=13) pyramidal neurons served as control group. All analyzed pyramidal neurons were in culture for at least 7 days. The corresponding data and statistics are presented in Figure 8 and revealed no significant differences for the analyzed parameters (resting membrane potential, input resistance, AP half width and sag potential).

In addition, we added an entirely new comparative analysis of GFP positive (n=6) and negative interneurons (n=9), that also did not show significant differences in most parameters, with the exception of the hyperpolarization induced sag potential, which was found to be reduced in GFP positive interneurons (corresponding data and statistics are presented in Figure 8).

Table 1 now provides additional information on all analyzed neurons regarding the surgery/patient they were derived from.

Although we increased the numbers of analyzed neurons and while we feel that our data clearly indicate no major interference of viral transduction with neuronal function, we understand that we are still dealing with limited sample sizes. Therefore, we added the following statements to the Discussion:

“Moreover, we measured and compared the properties of GFP positive neurons after viral transduction and found no major impairment of neuronal function in our analyzed samples of pyramidal cells (n=7) and interneurons (n=6), which is in line with a previous study investigating virus transduction in human neurons (Ting et al., 2018).”

“Although our data indicate no major interference of viral transduction with neuronal function we cannot rule out more subtle alterations, particularly given the immense diversity of cortical neurons, which certainly cannot be represented in its entirety in our samples. This question needs to be further approached in future large-scale studies and respective investigations will benefit from community-based efforts.”

2) The spine analysis is confusing. It's not clear what is being analyzed – is each data point in Figure 7 an averaged value of spine measurements per neuron? Where were the spine measurements sampled from in the neurons? The authors need to improve the clarity regarding how these experiments were conducted.

We analyzed the spines of biocytin filled and GFP labeled neurons as follows: For each pyramidal neuron we analyzed 5 areas of the dendritic tree (3 of the apical dendritic (red boxes) and 2 of the basal dendritic compartment (blue boxes), each 72 x 36 µm in size, please also see Figure 7). Spine density, spine length and spine head diameter were measured for each area. Values of all 5 areas per neuron were averaged and plotted against the “DIV” or against the “distance to pia” such that every data point in Figure 7E and F corresponds to the average of the 5 areas per neuron. We have chosen this method to minimize potential measuring errors for each analyzed neuron. For clarification we have merged former Figure 7 and Figure 8 to new Figure 7. The group data (former Figure 7E) were transferred to new Figure 7—figure supplement 1.

3) The Materials and methods are incomplete and would not allow a reader to repeat the experiment. For example, there is no mention of what media the slices were cultured in. Or how the slices were kept sterile, which seems like a key point.How were the slices treated after sectioning?Were they recovered in a warm incubation system as is convention? If so, what temperatures and times were used?Were the cultured slices sent directly to culture after slicing? After recovery? How were they treated? This are crucial points for this methods-oriented paper and the authors are extremely light on the details.When were electrophysiological recordings (and biocytin filling) of acute slices done after cutting? From experience of both human and rodent slice recordings, slices have to rest after slicing to recover due to processes like spine retraction and glycogen depletion (e.g. Kirov et al., 2004, Neuroscience), usually an hour for rodents and 3 hours for human hippocampal and cortical tissue. This should be described in Materials and methods.

We apologize for the incomplete description of our methods. We added the missing information to the Materials and methods section as follows:

“Preparation of slices from resected tissue: “Afterwards the slices were kept in aCSF equilibrated with carbogen for 0.5 h at room temperature before they were transferred either to patch setups for acute electrophysiological recordings and biocytin fillings or onto culture membranes for cultivation.”

Human brain slice cultures: “For the first hour following the slicing procedure the slices were cultured in 1.5 ml NSC media (48% DMEM/F-12 (Life Technologies), 48% Neurobasal (Life Technologies), 1x N-2 (Capricorn Scientific), 1x B-27 (Capricorn Scientific), 1x Glutamax (Life Technologies), 1x NEAA (Life Technologies) + 20 mM HEPES) before changing to 1.5 ml hCSF per well without any supplements. No antibiotics or antimycotics were used during cultivation.”

Whole-cell patch clamp recordings: “[…] continuously superfused with 32.0 +/- 1.5 °C aCSF (in mM: 118 NaCl, 25 NaHCO3, 30 D-glucose, 1 MgCl2, 3 KCl, 1 NaH2PO4, und 1.5 CaCl2) and visualized with a BX61WI Microscope (Olympus) or a Leica stereomicroscope. After 1 h of equilibration in the patch setup recordings were performed using recording electrodes with a resistance of 3-5 MΩ and filled with an intracellular whole-cell patch-clamp pipette solution containing”

4) The authors refer to the choline chloride solution they use to transport and slice the human tissue as aCSF, but they also refer to their patch clamp recording solution (which they do not define) as aCSF. It's highly unlikely they performed physiology experiments in the choline solution. Describing the constituency of the recording solution is necessary.

We apologize for the incomplete description of our aCSF solutions. We changed the respective passages as follows:

“[…] continuously superfused with 32.0 +/- 1.5 °C recording aCSF (in mM: 118 NaCl, 25 NaHCO_3_, 30 D-glucose, 1 MgCl2, 3 KCl, 1 NaH_2_PO_4_, und 1.5 CaCl_2_)”

5) Variation in patient material needs more attention.Patient material can be quite variable depending on various factors, such as the level of pathophysiology of the tissue, patient age and handling during surgery. In this manuscript there are 3 different cortical areas measured, the age range is from 10 to 56 and 2 patients presented with gangliogliomas. This can be a problem when samples sizes are low if the slices are all from the same patients and not representing the majority. The patient material underlying the analyses should be stated (with patients numbered) when describing groups and figures.

Table 1 now provides additional information on all analyzed neurons regarding the surgery/patient they were derived from.

6) The comparison between DIV 0 and DIV 2-14 for electrophysiology is a bit strange. If the goal is to investigate effect of time in culture should the comparison be between DIV 0/2 and DIV 14? The numbers of cells are not enough for this comparison.

We agree with the comments. We have added more neurons to the analysis and regrouped the data (Figure 1E) according to the reviewer’s suggestion as follows:

a) 0 DIV (n=17)

b) 2-3 DIV (n=8)

c) 7-14 DIV (n=20).

For these groups we performed Kruskal-Wallis tests with Dunn's multiple comparisons test to test for statistically significant differences.

Additionally we provide graphs plotting AP half width, resting membrane potential, sag potential and input resistance for each of the 45 analyzed pyramidal neurons as individual data points versus the DIV (Figure 1D).

We changed the Results section and Figure 1 accordingly:

“In a first set of experiments we determined the stability of the electrophysiological properties of pyramidal neurons performing whole cell patch clamp recordings (n=45) from cells in acute (n=17) or cultured (n=28) slices (2-14 DIV). […] In summary, increasing time in culture does not impact the majority of analyzed intrinsic electrophysiological characteristics of pyramidal neurons, except for the membrane resting potential which slightly changed to more depolarized values within the first 2-3 days in culture and then stayed stable over the remaining time.”

Additionally, we inserted the following paragraph in the Discussion:

“In the present study we now provide a systematic analysis of pyramidal neurons and interneurons across all cortical layers supporting overall electrophysiological and structural stability throughout the time window up to two weeks in culture. […] This might be interpreted as a small initial adaptation of pyramidal neurons to the new conditions in culture, particularly since we did not find continued depolarization of resting membrane potential in late stage cultures compared to early stage ones. Interestingly, interneurons did not seem to show this effect.”

On the same note the time-point for MAP2/Satb2-stainings were 9 DIV vs 14 DIV. If changes in neuronal number have occurred before 9 DIV this will not be detected and MAP2/Satb2 data would not support the statement of a robust survival of neurons throughout the culturing time.

We agree with the reviewer and added the additional time-point “0 DIV”. Corresponding data are presented in the Results section and in Figure 1—figure supplement 2. There were no significant differences between the 3 time points (0 DIV, 9 DIV and 14 DIV).

7) The study mainly focusses on pyramidal neurons. Although the Satb2/Map2 comparisons in Figure 1—figure supplement 2 do hint at stable presence of inhibitory cells in the culture, it would be a very valuable addition to include analysis on properties of inhibitory interneurons to the study. In the description of Figure 8, recordings from fast spiking neurons are mentioned, but not shown. Are morphological and/or physiological properties of interneurons equally well-maintained across the culturing period? Perhaps this can be addressed in the data the authors already have collected?

We agree that – although our focus was on pyramidal neurons – our study would benefit from an addition of interneuron data. Indeed, we had already started to collect preliminary data before the initial submission. Meanwhile, we were able to acquire additional data after the submission and during the current revisions so that we feel confident to include these data now as a new data figure (Figure 6). Interneurons are an even more diverse population than pyramidal neurons. While morphology and firing behavior are known to substantially differ between various subclasses of interneurons, there are some intrinsic properties (such as resting membrane potential, input resistance, AP half width and sag potential; Figure 6F, G) which can be considered comparatively more uniform and which therefore were found suitable for an assessment over time in culture. We recorded 22 interneurons, 7 in acute (0 DIV) and 15 in cultured slices (7-14 DIV). We found no significant differences in the properties measured between the groups (0 DIV) and cultured slices (7-14 DIV) (Figure 6G). Additionally, we provide graphs for the different intrinsic properties in relation to the DIV with each interneuron being represented as an individual data point (Figure 6F).

We amended the Results section accordingly:

“Electrophysiological properties of interneurons in human brain slice cultures versus acute slices

To further investigate beyond our earlier indirect assessment based on the ratio of NeuN and Satb2 double-positive neurons to all NeuN positive neurons, whether also interneurons survive in human brain slice cultures we first performed immunocytochemical stainings for calretinin. Unlike for other interneuron subtype identifiers, we were able to obtain reliable stainings for calretinin in acute and cultured human brain slice tissue. […] The firing behavior of the recorded interneurons showed distinct firing behaviors as described before for rodents and humans and ranged from fast spiking (Figure 6D) to none fast spiking firing patterns (Ascoli et al., 2008).”

8) It is stated that patched neurons were filled with the intracellular solution for a minimum of 10 minutes "allowing for full morphological reconstruction". My question is if 10 minutes is sufficient for diffusion of biocytin to reach distal dendrites? What would be the diffusion rate of biocytin in your setup? This could be estimated using the recording temperature, which is missing in Materials and methods.

We apologize for not having mentioned the recording temperature. Slices were continuously superfused with recording aCSF at 32.0 +/- 1.5 °C (we added this information in the Materials and methods subsection “Whole-cell patch clamp recordings”). For recording temperatures on the order of the one we used a minimum recording/filling time of 10 minutes is in agreement with the literature (Karadottir and Attwell, Nature Protocols, 2006; Swietek et al., J Vis Exp., 2016). Though we stated that we filled the neurons for a minimum of 10 minutes, on average electrophysiological recordings took longer (20 – 30 minutes).

References: Káradóttir, R., and Attwell, D. (2006). Combining patch-clamping of cells in brain slices with immunocytochemical labeling to define cell type and developmental stage. Nature Protocols. https://doi.org/10.1038/nprot.2006.261

Swietek, B., Gupta, A., Proddutur, A., and Santhakumar, V. (2016). Immunostaining of biocytin-filled and processed sections for neurochemical markers. Journal of Visualized Experiments. https://doi.org/10.3791/54880

9) Related to Figure 1G: If brain slice properties alter over culture time, for instance connectivity or cellular morphology, this could show up in input resistance or membrane time constant. Were these parameters affected by days in culture? Please also test for potential correlations between input resistance and membrane time constant change with amount of time in culture.

We agree with the reviewer that additionally to the intrinsic parameters already analyzed by us (AP half width, resting membrane potential and sag potential) changes of the neuronal morphology and viability could also be reflected by changes of passive membrane properties such as input resistance and membrane time constant. As suggested we added an analysis of the input resistance of neurons recorded at different time points in culture. No significant differences between neurons within acute slices or within slices cultured for 2-3 days or 7-14 days were found. Furthermore, data did not reveal a positive or negative correlation of input resistance with the number of days in vitro. As demonstrated by Isokawa et al., Brain Res Brain Res Protoc., 1997 membrane time constant can be applied to indirectly estimate dendritic degeneration of electrotonically compact hippocampal dentate granule cells. However, such an indirect assessment is unable to provide detailed insight in the actual morphology of dendritic arbors and emanating spines, as required in our study. We therefore performed direct morphological analyses based on biocytin fillings and virus mediated labeling with GFP. Figure 1 was rearranged and data for AP half width, resting membrane potential, sag potential and input resistance are now presented in Figure 1D as individual data points per neuron versus the DIV and as group analysis (Figure 1E) with each neuron having been assigned to one of three culture time periods: acute (0 DIV), early (2-3 DIV) or late (7-14 DIV). We changed the Results section accordingly: “In a first set of experiments we determined the stability of the electrophysiological properties of pyramidal neurons performing whole cell patch clamp recordings (n=45) from cells in acute (n=17) or cultured (n=28) slices (2-14 DIV). […] In summary, increasing time in culture does not impact the majority of analyzed intrinsic electrophysiological characteristics of pyramidal neurons, except for the membrane resting potential which slightly changed to more depolarized values within the first 2-3 days in culture and then stayed stable over the remaining time.”

Reference: Isokawa, M. (1997). Membrane time constant as a tool to assess cell degeneration. Brain Research Protocols. https://doi.org/10.1016/S1385-299X(96)00016-5

10) From culturing rodent cortical brain slices it is known that slices flatten out during culture period. Is the three dimensional structure maintained during 3 weeks of adult human slice culture? Does slice thickness decrease over 3 weeks culturing? Please provide numbers.

As pointed out by the reviewer rodent cortical brain slices flatten out quite extensively in culture. Though we see some flattening of human slice cultures, this effect is a lot less pronounced than for rodent tissue. We measured the thickness of the slices over several time points and provide the data in Figure 1—figure supplement 1D.

11) In Figure 3 neurons were assigned to cortical layers based on depth from pia. Were layer dimensions stable over the entire culture period? This could be determined by Nissl stains at different DIVs. Please provide numbers.

We thank the reviewer for this important comment. For a validation of stability of layer dimensions throughout the culturing time slices were analyzed for different time points in culture based on DAPI staining. Layer dimensions were found to remain robust during culturing. The corresponding data are now included in Figure 1—figure supplement 1B, C, E.

12) Figure 4: reconstructing GFP positive neurons may be hampered by overlapping processes containing GFP from neighboring neurons. Can the authors rule out that 'contamination' of dendritic structure reconstructions by overlap of stained processes from different individual neurons affected their quantification?13) In Figure 5C, right panel, the comparisons of the total dendritic length show a slight difference between the reconstructed biocytin and GFP labelled cells. The authors mention that 'very distal dendritic ramifications being captured less reliably'. It is not clear to me what would be the cause of this reduced reliability. Please explain.

We agree that overlapping GFP positive processes from neighboring neurons make a reconstruction more challenging then single biocytin filled neuron reconstructions. This is what we meant by the statement “very distal dendritic ramifications being captured less reliably”. Since tracing and reconstruction of dendritic processes is not performed in 2D maximum projection images but in 3D rotatable z-stack volumes, we are convinced that the reconstructions we are providing based on GFP are accurate, with the minor exception of some uncertainties when approaching dendritic terminal endpoints. We modified the corresponding paragraph in the Results section:”Comparing combined total dendritic length of all biocytin and GFP labeled neurons independent of the DIV revealed a slight underestimation of values by GFP (Figure 5C, right panel, p<0.01, n=24 (biocytin), n=23 (GFP), unpaired Mann Whitney test) – likely because very distal dendritic ramifications are being captured somewhat less reliably (due to intermingling GFP positive processes from neighboring neurons and some arising uncertainty when judging whether to assign GFP positive processes in the periphery of the neuron being reconstructed to the filament of this neuron or whether to discard them by deeming them as originating from neighboring neurons) – but overall successful reconstruction of the major parts of dendritic arbors (Figure 2 and Figure 4).”

For illustration we uploaded all reconstructed neurons to the Dryad server in 3D rotatable format.

14) In Figure 8: please provide quantitative support of the statements that physiological properties and AP firing of GFP positive neurons was 'normal'.

Please also refer to the response to reviewer comment 1. We now provide quantitative data along with statistical analysis and present the corresponding data in Figure 8. The number of analyzed pyramidal neurons after virus transduction was doubled from n=3 to n=7. Non-transduced GFP negative (n=13) pyramidal neurons served as control group. All analyzed pyramidal neurons were in culture for at least 7 days. The corresponding data and statistics are presented in Figure 8 and revealed no significant differences for the analyzed parameters (resting membrane potential, input resistance, AP half width and sag potential).

In addition, we added an entirely new comparative analysis of GFP positive (n=6) and negative interneurons (n=9), that also did not show significant differences in most parameters, with the exception of the hyperpolarization induced sag potential, which was found to be reduced in GFP positive interneurons (corresponding data and statistics are presented in Figure 8).

The Results section was modified accordingly: “To investigate whether AAV-transduction and GFP expression affected the electrophysiological properties of cortical neurons, we performed whole-cell patch clamp recordings of GFP positive neurons (n=7) and GFP negative neurons (n=13) recorded at matching time points in culture (7-16 DIV) and assessed AP half width, resting membrane potential, sag potential and input resistance. […] Taking together these results, virus transduction did not dramatically interfere with neuronal functionality.”

15) Were synaptic current frequencies and amplitudes comparable to GFP-negative neurons at different DIVs?

We agree with the reviewer that synaptic currents are of great interest in human slices cultures. We measured occasionally the synaptic currents in voltage clamp, but focused in this study on intrinsic firing properties of the neurons and morphology. We feel that a thorough analysis of synaptic transmission would exceed the scope of this study and should be carefully performed in future studies. We removed the example traces of the GFP positive and negative neurons.

16) In the final paragraph of the Results section, the authors hint at possibility to follow spine morphology changes in time, possibly across days. For this it is required that the authors succeed in imaging the same stretch of dendrite across several days. It would be a nice addition to the study if the authors could provide data to support this possibility.

We agree with the reviewer, that the imaging of dendrites and subsequently spines could be a very useful tool for many basic and disease related questions. We therefore performed a first set of experiments imaging several areas of a slice culture over the time period of 24 hours (Figure 9 and Figure 9—figure supplement 1). We analyzed the stability of the dendritic length and maximum sholl radius over this time period and showed only minimal changes of these parameters. The following section was added to the Results section:

“In a second step we asked whether our system would be stable enough to enable time-lapse tracing of dendrites and spines of human pyramidal neurons over a time window of several hours. […] After further optimization this approach will enable future studies of spine turnover of individual human neurons within brain slice cultures.”